# Tbx5 drives *Aldh1a2* expression to regulate a RA-Hedgehog-Wnt gene regulatory network coordinating cardiopulmonary development

Scott A Rankin[1], Jeffrey D Steimle[2,3,4†], Xinan H Yang[2,3,4], Ariel B Rydeen[2,3,4], Kunal Agarwal[1], Praneet Chaturvedi[1], Kohta Ikegami[2‡], Michael J Herriges[2], Ivan P Moskowitz[2,3,4]*, Aaron M Zorn[1,5]*

[1]Center for Stem Cell and Organoid Medicine (CuSTOM), Division of Developmental Biology, Perinatal Institute, Cincinnati Children's Hospital Medical Center, Cincinnati, United States; [2]Department of Pediatrics, University of Chicago, Chicago, United States; [3]Department of Pathology, University of Chicago, Chicago, United States; [4]Department of Human Genetics, University of Chicago, Chicago, United States; [5]University of Cincinnati, College of Medicine, Department of Pediatrics, Chicago, United States

*For correspondence:
imoskowitz@peds.bsd.uchicago.edu (IPM);
aaron.zorn@cchmc.org (AMZ)

Present address: †Department of Molecular Physiology and Biophysics, Baylor College of Medicine, Houston, United States; ‡Division of Molecular and Cardiovascular Biology, Heart Institute, Cincinnati Children's Hospital Medical Center, Cincinnati, United States

Competing interest: The authors declare that no competing interests exist.

**Abstract** The gene regulatory networks that coordinate the development of the cardiac and pulmonary systems are essential for terrestrial life but poorly understood. The T-box transcription factor Tbx5 is critical for both pulmonary specification and heart development, but how these activities are mechanistically integrated remains unclear. Here using *Xenopus* and mouse embryos, we establish molecular links between Tbx5 and retinoic acid (RA) signaling in the mesoderm and between RA signaling and sonic hedgehog expression in the endoderm to unveil a conserved RA-Hedgehog-Wnt signaling cascade coordinating cardiopulmonary (CP) development. We demonstrate that Tbx5 directly maintains expression of *aldh1a2*, the RA-synthesizing enzyme, in the foregut lateral plate mesoderm via an evolutionarily conserved intronic enhancer. Tbx5 promotes posterior second heart field identity in a positive feedback loop with RA, antagonizing a Fgf8-Cyp regulatory module to restrict FGF activity to the anterior. We find that Tbx5/Aldh1a2-dependent RA signaling directly activates *shh* transcription in the adjacent foregut endoderm through a conserved MACS1 enhancer. Hedgehog signaling coordinates with Tbx5 in the mesoderm to activate expression of *wnt2/2b*, which induces pulmonary fate in the foregut endoderm. These results provide mechanistic insight into the interrelationship between heart and lung development informing CP evolution and birth defects.

## Introduction

Proper integration of the cardiac and pulmonary systems begins during early embryogenesis and is essential for terrestrial life. A key feature of cardiopulmonary (CP) development is evolutionarily conserved bi-directional paracrine signaling between the foregut endoderm, which gives rise to pulmonary epithelium, and the cardiogenic mesoderm (*Xie et al., 2012*; *Rankin et al., 2016*; *Steimle et al., 2018*). The interplay between these signals and lineage-specific transcription factors (TFs) to control lineage-specific gene regulatory networks (GRNs) for heart and lung morphogenesis is poorly understood. A better understanding of these CP GRNs will provide insight into the orchestration of heart and lung development and inform the molecular basis of life-threatening CP birth defects.

The vertebrate heart forms from two distinct populations of cardiac progenitor cells in the anterior lateral plate mesoderm (lpm), termed the first and second heart fields, respectively (FHF and SHF; *Kelly et al., 2014*). The FHF differentiates first and forms the early heart tube, including portions of the two atria and left ventricle. The SHF contributes to the anterior and posterior poles of the developing heart and differentiates later. The anterior SHF (aSHF) is characterized by the expression of *Fgf8, Fgf10,* and *Tbx1* and generates the right ventricle, portions of the outflow tract, and pharyngeal mesoderm (*Rochais et al., 2009*; *Kelly et al., 2014*). The posterior SHF (pSHF) is characterized by the expression of *Tbx5, Osr1,* and *Foxf1,* (*Xie et al., 2012*; *Hoffmann et al., 2014*; *Steimle et al., 2018*) and generates the atrial septum and sinus venosus. A subset of the pSHF marked by *Isl1, Gli1,* and *Wnt2* expression contains multipotent cardiopulmonary progenitors (CPPs) that give rise to lung mesenchyme, pulmonary vasculature, and myocardium of the inflow tract (*Peng et al., 2013*). CPPs are both the recipient and source of reciprocal signaling with the adjacent pulmonary endoderm essential for heart and lung development.

The T-box TF Tbx5 is a key player coordinating CP organogenesis. Numerous studies in vertebrate animal models over the past 20 years have documented conserved *Tbx5* expression initially in the FHF and then later in the pSHF. Heterozygous mutations in human*TBX5* cause Holt–Oram syndrome with congenital heart anomalies including atrial septal defects and hypoplastic left heart (*Li et al., 1997*; *Ryan and Chin, 2003*). *Tbx5*$^{-/-}$ null mutant mice die between E9.5 and E10.5 with severe cardiac deficiencies and a failure of pulmonary development (*Bruneau et al., 2001*; *Xie et al., 2012*; *Hoffmann et al., 2014*; *Steimle et al., 2018*; *De Bono et al., 2018*). While significant advances have been made in understanding the TBX5-regulated GRNs controlling cardiomyocyte development (*Kathiriya et al., 2021*), how TBX5 coordinates heart and lung organogenesis is less clear. We recently showed that Tbx5 is non-cell-autonomously required to activate expression of Hedgehog (Hh) ligands in the adjacent foregut endoderm, which are essential for both heart and lung development (*Steimle et al., 2018*). Endodermal Hh signals back to the lpm stimulating Gli TFs, which cooperate with Tbx5 to directly activate expression of mesodermal *Wnt2/2b* signals that are essential to induce pulmonary fate in the adjacent foregut endoderm (*Steimle et al., 2018*; *Goss et al., 2009*; *Harris-Johnson et al., 2009*). Tbx5 is thereby required for establishing the reciprocal mesoderm—endoderm—mesoderm signaling loop that coordinates CP development. A major unanswered question is how Tbx5 non-cell-autonomously activates sonic hedgehog (Shh) ligand expression in the foregut endoderm.

Retinoic acid (RA) signaling is a strong candidate for the Tbx5-dependent signal that activates endodermal *Shh* expression. Like *Tbx5* mutants, RA deficient embryos have reduced *Shh* expression in the foregut (*Wang et al., 2006*; *Rankin et al., 2016*) and manifest multiple cardiac and pulmonary defects, like Tbx5 mutants (*Zaffran et al., 2014*; *Xavier-Neto et al., 2015*; *Perl and Waxman, 2019*; *Sirbu et al., 2020*). RA, a derivative of vitamin A, is produced in the lpm by the aldehyde dehydrogenase enzyme Aldh1a2, which converts cellular retinaldehyde into RA (*Niederreither et al., 1999*; *Metzler and Sandell, 2016*). *Aldh1a2* and *Tbx5* are co-expressed in a subset of the pSHF and previous studies have shown that RA patterns the SHF by promoting *Tbx5+* pSHF identity whilst repressing *Tbx1*$^+$ aSHF fate (*Niederreither et al., 2001*; *Sirbu et al., 2008*; *Ryckebusch et al., 2008*; *Deimling and Drysdale, 2009*; *Ryckebüsch et al., 2010*; *Rydeen and Waxman, 2016*; *Rankin et al., 2016*; *De Bono et al., 2018*). How the regional production of RA is controlled to pattern the SHF and regulate *shh* expression remains unknown.

In this study, we demonstrate that RA signaling is the link between mesodermal Tbx5 activity and endodermal *Shh* expression. We further define the molecular basis by which Tbx5 drives RA signaling and by which RA signaling drives *Shh* expression. Specifically, Tbx5 directly maintains expression of *Aldh1a2* in pSHF via an evolutionarily conserved intronic enhancer, and Tbx5/Aldh1a2-dependent RA signaling directly activates *Shh* transcription in the foregut endoderm via an evolutionarily conserved MACS1 endoderm enhancer. We conclude that Tbx5 coordinates CP development by controlling expression of the RA-producing enzyme *Aldh1a2*, and that this RA signal initiates a mesenchyme-epithelial signaling cascade that controls both Hh/Wnt-dependent lung induction and SHF patterning. Hh/Gli and Tbx5 then cooperate to promote Wnt2/2b expression and lung induction. This work unifies previously unconnected observations to resolve the molecular basis of a mesoderm-endoderm-mesoderm signaling network that coordinates pulmonary induction and SHF cardiac patterning.

# Results

## Tbx5 regulates cardiopulmonary development and *Aldh1a2* expression in mouse

To investigate the Tbx5-regulated GRN that coordinates heart and lung development, we re-examined our published RNA-seq data of CP tissue (containing both foregut mesoderm and endoderm) micro-dissected from wild-type (WT) and *Tbx5*⁻/⁻ mouse embryos at E9.5 (*Steimle et al., 2018*). Differential expression analysis revealed 1588 upregulated genes and 1480 downregulated genes and in the absence of *Tbx5* (≥1.5 fold change and 5% FDR) (*Figure 1A*; *Figure 1—source data 1*; *Steimle et al., 2018*). Reduced expression of Hh signaling components (*Shh*, *Ihh*, and *Patch2*), Hh-targets (*Hhip* and *Gli1*), the lung-inducing *Wnt2/2b* ligands, and pulmonary progenitor marker *Nkx2-1*, indicated a loss of pulmonary fate in *Tbx5*⁻/⁻ mutant CPP tissue (*Figure 1B*). We examined the relationship between pSHF/lung and aSHF/pharyngeal transcripts in *Tbx5*⁻/⁻ embryos by intersecting the Tbx5-regulated transcriptome with gene sets from recent single-cell RNA-seq studies of the developing E7.75–E9.5 mouse heart and foregut that define aSHF, pSHF, pharynx, and lung progenitor cells (*de Soysa et al., 2019*; *Han et al., 2020*, *Supplementary files 2 and 3*). We found that 25% of genes (91/366) specifically enriched in aSHF or pharyngeal cells, but only 5% of genes (10/213) specifically enriched in pSHF or lung progenitors, overlapped with transcripts upregulated in the *Tbx5*⁻/⁻ mutants (*p<0.0001, hypergeometric probability test, HGT) (*Figure 1A*). On the other hand, 34% of the pSHF or lung marker genes (72/213) but only 6% of the pSHF+ pharynx enriched genes (21/366) were downregulated in *Tbx5*⁻/⁻ mutants (*p<0.001, HGT) (*Figure 1A*). The aSHF-enriched genes upregulated in *Tbx5*⁻/⁻ CP tissue included well-known patterning genes *Hand1*, *Irx3*, *Irx5*, *Mef2c*, *Meg3*, *and Tlx1* as well as FGF signaling components and targets including *Fgf8*, *Fgf10*, *Spry1*, *Spry2*, and *Dusp6* (*Figure 1B*). Gene set enrichment analysis (GSEA) confirmed a statistically significant overrepresentation of aSHF/pharynx genes among upregulated genes (normalized enrichment score [NES]=1.58; p<0.0001) and overrepresentation of pSHF/lung transcripts among the downregulated genes (NES=–1.99; p<0.0001) in the *Tbx5*⁻/⁻ CP tissue (*Figure 1—figure supplement 1A,B*). Thus, *Tbx5* mutant mouse embryos exhibit a reduction of the pSHF transcriptional program and gain of aSHF gene expression in the pSHF domain, consistent with recent reports (*De Bono et al., 2018*).

These changes in pSHF gene expression suggested a possible loss of RA signaling. Indeed, genes that promote RA signaling were downregulated in *Tbx5*⁻/⁻ CP tissue, including *Aldh1a2*, *Crabp2*, which promotes nuclear shuttling of RA, and *Rbp1*, a cytosolic chaperone of the RA precursor retinol (*Figure 1B*). On the other hand, enzymes that attenuate RA signaling, including *Cyp26a1*, *Cyp26b1*, *Cyp26c1*, and *Dhrs3*, were increased in *Tbx5*⁻/⁻ pSHF/CPP tissue. Reduced RA-signaling in *Tbx5*⁻/⁻ CP tissue was also consistent with increased expression of multiple TGFβ pathway components and targets (*Figure 1—figure supplement 1C*), which are known to be suppressed by RA during foregut and heart development (*Chen et al., 2007*; *Li et al., 2010*; *Ma et al., 2016*).

The observation that *Aldh1a2* expression was reduced in the *Tbx5*⁻/⁻ pSHF whereas FGF signaling components and targets were increased is consistent with the known role of RA in negatively regulating *Fgf8/Fgf10+* aSHF fate (*Ryckebusch et al., 2008*; *Sirbu et al., 2008*; *Rydeen and Waxman, 2016*). RT-qPCR of dissected E9.5 CP tissue validated the RNA-seq analysis with *Aldh1a2* being dramatically downregulated in *Tbx5*⁻/⁻ mutants while *Fgf8* and *Fgf10* were upregulated (*Figure 1C*). Immunostaining of transgenic *Shh:GFP* embryos confirmed previous reports that Aldh1a2 is co-expressed with Tbx5 in a subset of pSHF cells adjacent to the foregut Shh/Nkx2-1-expressing pulmonary endoderm (*Figure 1D–F"* and *Figure 1—figure supplement 1D-E'' Hochgreb et al., 2003*; *Ryckebusch et al., 2008*; *De Bono et al., 2018*; *de Soysa et al., 2019*). Co-expression of *Tbx5* and *Aldh1a2* transcripts in the pSHF adjacent to *Tbx1*⁺ aSHF and *Shh* + endoderm were also evident in an online spatial atlas of single-cell gene expression (*Figure 1—figure supplement 1F*; *Lohoff et al., 2021*). To further define the spatial distinct aSHF-FGF and pSHF-RA signaling domains, we generated 3D reconstructions using serial sections of *Aldh1a2*, *Fgf8*, *Fgf10,* and *Shh* in-situ hybridizations from the WT mouse E9.5 (*Figure 1—figure supplement 1G-H*). Together these observations were consistent with the hypothesis that Tbx5 regulates CP development by controlling expression of *Aldh1a2*, which in turn establishes a local domain of RA activity in the pSHF that suppresses aSHF fate and promotes pulmonary development.

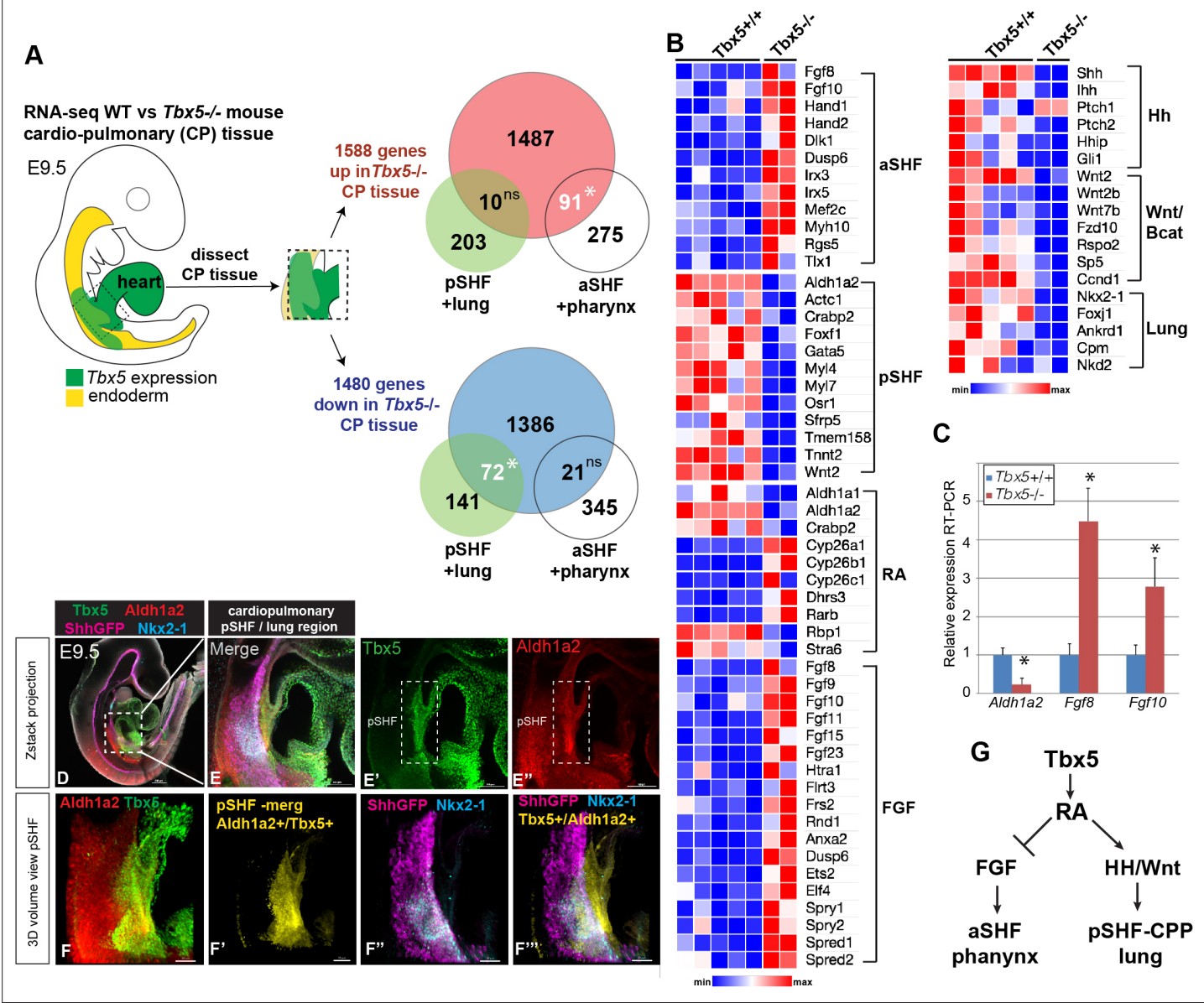

**Figure 1.** Tbx5 is required for posterior cardiopulmonary (CP) development. (**A**) Schematic of an E9.5 mouse embryo highlighting the dissected CP tissue (containing foregut mesoderm and endoderm) profiled by bulk RNA-seq. Venn diagrams show genes differentially expressed in wild-type (WT) versus *Tbx5⁻/⁻* CP tissue (>1.5 fold change, 5% FDR; *Steimle et al., 2018*, GSE75077) intersected with gene sets from single-cell RNA-seq studies defining aSHF+ pharynx cells versus pSHF/CPP+ lung progenitor cells (*de Soysa et al., 2019*, GSE126128; *Han et al., 2020*, GSE136689) (*Supplementary files 2 and 3*). Statistically significant intersection based on hypergeometric tests. *p<0.0001. (**B**) Transcriptome analysis of *Tbx5⁻/⁻* CP tissue suggests disrupted SHF patterning and failed pulmonary development with reduced RA and increased FGF signaling. Heat map of selected differentially expressed genes in WT *Tbx5⁺/⁺* (n=5) and *Tbx5⁻/⁻* mutant (n=2) CP tissue grouped by domain of expression or pathway. (**C**) RT-qPCR validation of decreased *Aldh1a2* and increased *Fgf8, Fgf10* expression in E9.5 WT and *Tbx5⁻/⁻* CP tissue. Relative mean expression+S.D. *p<0.05 Student's t-test relative to WT littermates. (**D–F**) Whole-mount immunostaining of E9.5 *Shh:GFP* mouse embryos show that Tbx5 (green) and Ald1a2 (red) expression overlaps in a subset of the pSHF (yellow in (**F′**) and (**F‴**)) adjacent to the Nkx2-1+/Shh:GFP+ pulmonary domain (**F″, F‴**). Scale bar in (**D**) = 200 μM, (**E–E″**) = 100 μM, and (**F–F‴**) = 50 μM. (**G**) Model of the proposed Tbx5-RA signaling networks in the cardiopulmonary tissue. Also see *Figure 1—figure supplement 1*, *Figure 1—figure supplement 1*– tables 1-3. aSHF, anterior second heart field; fg endo, foregut endoderm; ns, not significant; oft, outflow tract; pSHF, posterior SHF; RA, retinoic acid; ventr, ventricle.

The online version of this article includes the following figure supplement(s) for figure 1:

**Source data 1.** Differentially expressed genes in mouse E9.5 micro-dissected cardiopulmonary progenitor (CPP) tissue based on bulk RNA-seq (*Steimle et al., 2018*, GSE GSE75077).

*Figure 1 continued on next page*

*Figure 1 continued*

**Figure supplement 1.** Analysis of the Tbx5 regulated transcriptome in mouse cardiopulmonary tissue and co-expression of Tbx5 with Aldh1a2 in the pSHF.

## Tbx5 regulates cardiopulmonary development and maintains *Aldh1a2* expression in *Xenopus*

Since *Tbx5⁻/⁻* mutant mouse embryos die shortly after E9.5 from cardiac insufficiency (***Bruneau et al., 2001***; ***Xie et al., 2012***), we turned to *Xenopus* to elucidate the molecular mechanisms by which Tbx5 coordinates CP development. *Xenopus* larva can live for many days without a functional heart, by absorbing oxygen from the water, and their experimental advantages facilitate epistatic analysis of signaling pathways.

Previous studies have shown that Tbx5-regulated CP development is conserved between *Xenopus* and mouse: Tbx5 loss-of-function (LOF) in *Xenopus*, either by CRISPR/CAS9-mediated mutation or morpholino (MO) knockdown, phenocopies the mouse *Tbx5⁻/⁻* phenotype with severe cardiac hypoplasia, a failure to induce Nkx2-1+ lung progenitors and the foregut tube fails to separate into distinct trachea and esophagus (***Steimle et al., 2018***; ***Brown et al., 2005***; ***Figure 2—figure supplement 1A***).

Analysis of control and Tbx5 depleted *Xenopus* embryos showed that, like in mice, Tbx5 and Aldh1a2 were co-expressed in the foregut lpm/pSHF (***Figure 2A and B***) and that *Xenopus* Tbx5 is required for *aldh1a2* expression (***Figure 2C–F***). Both *X. laevis* Tbx5-MO morphant and *X. trop tbx5* CRISPR/CAS9 mutant embryos exhibited a loss or strong reduction of *aldh1a2* transcripts and Aldh1a2 protein in the foregut lpm at NF34 (a timepoint similar to mouse E9.5) (***Figure 2C–F***; ***Figure 2—figure supplement 1B***). Quantification of the Aldh1a2 immunostaining in 3D volume renderings of the fg lpm/pSHF domain of Tbx5 morphants and mutants revealed that Aldh1a2 protein was only expressed on average to approximately  28% (p=0.0009) and  33% (p≤0.0001) of WT levels, respectively (***Figure 2—figure supplement 1B*** and ***Figure 3K***). Analysis of transgenic Wnt/β-catenin reporter embryos *Tg(WntRE:dGFP)* (***Tran et al., 2010***), confirmed the failure of Wnt-dependent pulmonary induction in the ventral foregut of Tbx5-deficient embryos (***Figure 2C and D***). Importantly, co-injection of human *TBX5* RNA rescued *aldh1a2* expression and pulmonary development (***Figure 2D–F***, ***Figure 2—figure supplement 1B***, and ***Figure 3***). A time course analysis revealed that loss of Tbx5 resulted in a downregulation of *aldh1a2* expression in the foregut lpm starting at NF25, but not at early somitogenesis stages (NF15) (***Figure 2E and F***). These results demonstrate that Tbx5 is required to maintain *aldh1a2* expression in the foregut lpm, and that it regulates a conserved transcriptional program in *Xenopus* and mouse to coordinate SHF patterning and lung induction.

## Tbx5 regulates cardiopulmonary development via RA signaling

A detailed analysis of Tbx5-MO embryos by in-situ hybridization and RT-PCR showed that many of the CP genes that were misregulated in mouse *Tbx5⁻/⁻* CP tissue were also misregulated in *Xenopus* (***Figure 3A–K***). In addition to a loss of *aldh1a2*, pSHF markers *osr1*, *foxf1*, *gli1*, and *wnt2b* and pulmonary endoderm markers *shh*, *dhh*, and *nkx2-1* were reduced, while pharyngeal/aSHF markers *fgf8*, *fgf10*, *tbx1*, *cyp26a1*, *cyp26c1*, *spry2*, *hand1*, *hand2*, *dhrs3*, *tgfbR2,* and *tgfbi* were all upregulated. In total, all 19 transcripts tested exhibited changes in gene expression similar to *Tbx5⁻/⁻* mice. Interestingly in the Tbx5-MO embryos, we observed changes in gene expression beyond just the CP region, including the kidney, pharynx, and head all of which are known to be regulated by RA, FGF, and/or Hh signaling. This suggests that non-cell-autonomous effects in Tbx5 depleted embryos are likely due to changes in secreted factors.

We hypothesized that the disrupted CP development in Tbx5-deficient *Xenopus* embryos was primarily caused by reduced Aldh1a2-dependent RA signaling. To address this, we tested whether blocking endogenous RA could phenocopy loss of Tbx5 or if addition of exogenous RA could rescue the Tbx5 LOF phenotype (***Figure 3A***). We suppressed endogenous RA synthesis by addition of the Aldh enzyme inhibitor DEAB between NF20 and NF34, the time when *aldh1a2* expression was Tbx5-dependent. This phenocopied the Tbx5 LOF with loss of pSHF and pulmonary markers and an expansion of aSHF gene expression (***Figure 3B–J***). While DEAB allowed temporal-specific inhibition, in some instances pharmacological reagents can have off-target effects. Therefore, we also depleted

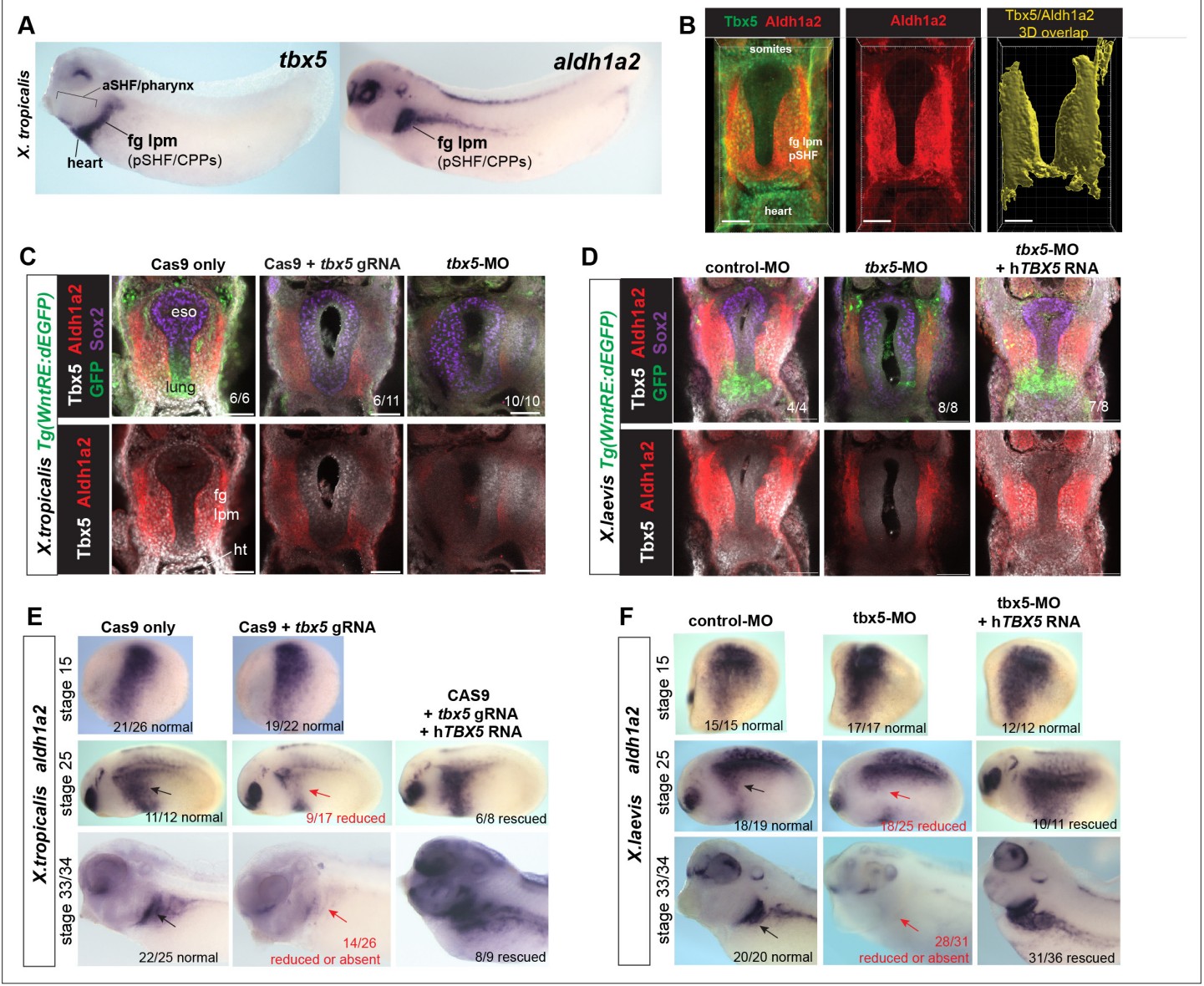

**Figure 2.** Tbx5 maintains *aldh1a2* expression in *Xenopus* foregut lpm.

The online version of this article includes the following source data and figure supplement(s) for figure 2:

**Source data 1.** Summary of in-situ hybridization results from *Xenopus* CAS9/CRISPR injection experiments in F0 embryos.

**Figure supplement 1.** Tbx5 LOF phenotype at stage NF43/44 and quantification of reduced Aldh1a2 immunostaining.

**Figure supplement 1—source data 1.** Quantitation of *Xenopus* NF34 fg lpm/pSHF Aldh1a2 immunostaining volume pixel intensity.

Aldh1a2 protein by targeted microinjection of MOs into the foregut region, which phenocopied the DEAB treatment (*Figure 3—figure supplement 1A-E*).

Importantly, we could partially rescue the Tbx5-MO, Aldh1a2-MO, and DEAB phenotypes with exogenous RA between NF20 and NF34, using a physiological concentration of 25 nM (*Horton and Maden, 1995*; *Mic et al., 2003*; *Sheikh et al., 2014*; *Figure 3B–J*; *Figure 3—figure supplement 1E*). Addition of RA suppressed the expanded expression domains of aSHF markers *fgf8, fgf10, spry2, cyp26a1, cyp26c1, and tbx1* in Tbx5/RA-deficient embryos (*Figure 3B–J*; *Figure 3—figure supplement 1E*). RA was also sufficient to rescue endodermal expression of *shh* and *dhh* as well as expression of known Hh-target genes *gli1, foxf1, and osr1* in the foregut lpm of Tbx5-MO embryos and explants (*Figure 3B–K*). However, exogenous RA did not rescue expression of the pulmonary-inducing *wnt2/2b* ligands nor the lung marker *nkx2-1*. In contrast, addition of recombinant WNT2B

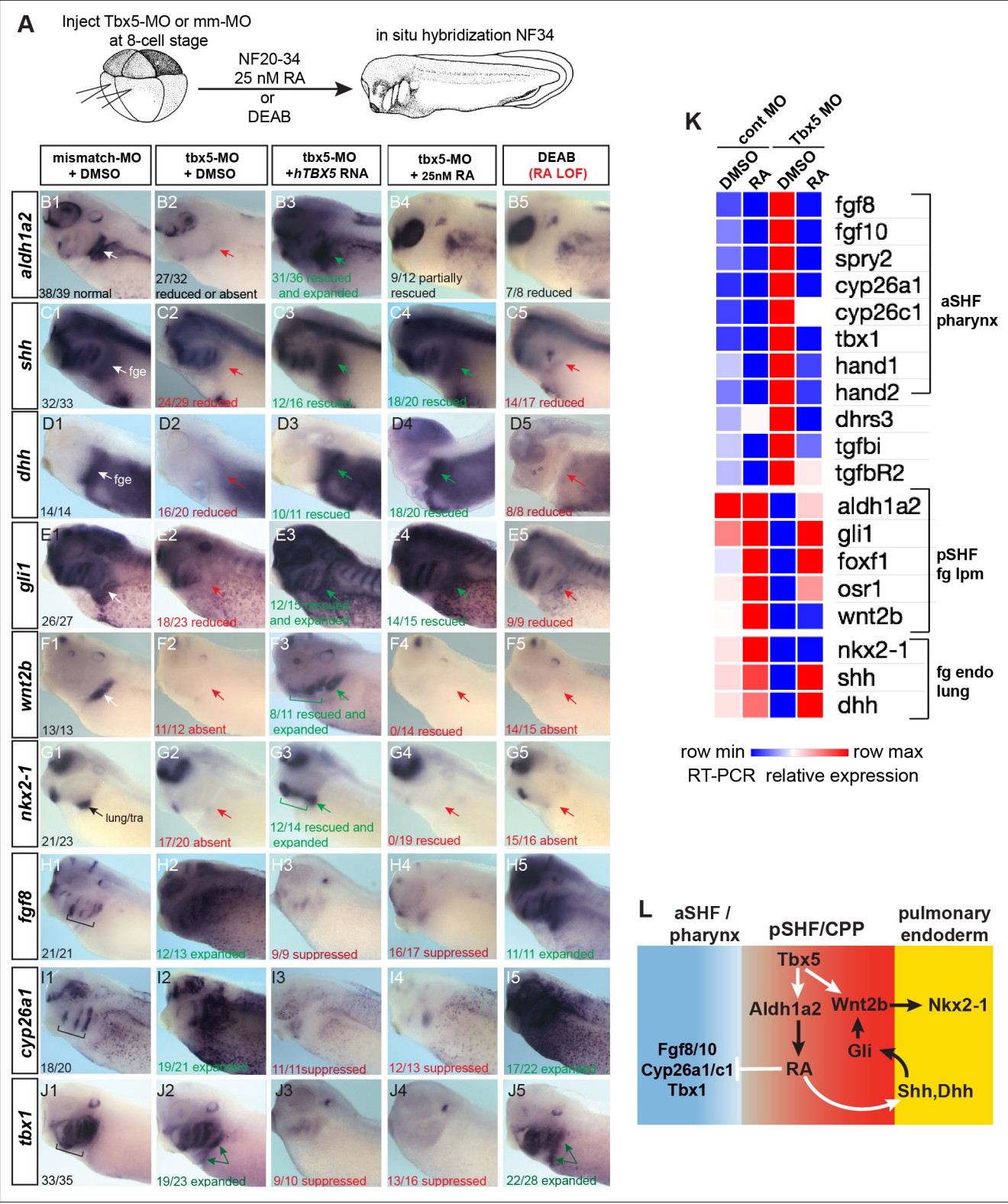

**Figure 3.** Tbx5 regulates *Xenopus* cardiopulmonary development in part via RA. (**A**) Schematic of the experimental design. (**B–J**) Exogenous RA rescues Tbx5 LOF, while inhibition of RA phenocopies Tbx5 LOF. Whole-mount in-situ hybridization of NF34 *X. laevis embryos* after the indicated experimental treatments: injection of negative control 3 bp mismatch-MO (10 ng), Tbx5-MO (10 ng), Human *TBX5* RNA (*hTBX5;* 100 pg), and/or 25 nM RA, 10 μM DEAB, DMSO vehicle control from NF20-34. The numbers of embryos with the observed expression pattern are indicated. Arrows indicate the relevant expression domain in the cardiopulmonar (CP) tissue. Brackets indicate the aSHF/pharyngeal domain. (**K**) Heat map showing relative expression from

*Figure 3 continued on next page*

*Figure 3 continued*

RT-PCR analysis of NF34 CP-foregut (fg) tissue dissected from control or Tbx5-MO injected embryos and treated with or without RA from NF20 to NF34. Each row is the average from the three biological replicates (n=4 explants per replicate). (**L**) Diagram of the proposed GRN model at NF25–35 showing the key role of Aldh1a2-dependent RA signaling downstream of Tbx5. White arrows indicate relationships tested in the above experiments and black arrows are demonstrated from the previous publications. Also see *Figure 3—figure supplement 1*, and related source data files. GRN, gene regulatory network; LOF, loss-of-function; MO, morpholino; RA, retinoic acid.

The online version of this article includes the following figure supplement(s) for figure 3:

**Source data 1.** *Figure 3K*.

**Figure supplement 1.** Aldh1a2 morpholino (MO) knockdown phenocopies DEAB treatment and WNT2B protein rescues *Nkx2-1+* pulmonary fate.

**Figure supplement 1—source data 1.** *Xenopus* explant RT-qPCR source data.

protein to Tbx5-MO foregut explants was sufficient to rescue *nkx2-1+* lung fate but not *shh* nor *dhh* expression (*Figure 3—figure supplement 1F,G*), consistent with previous reports that Tbx5 directly promotes *wnt2/2b* transcription (*Steimle et al., 2018*).

These results combined with our previous data suggest that Tbx5 promotes CP development by multiple mechanisms, which are experimentally separable (*Figure 3L*). First, by maintaining *aldh1a2* expression, Tbx5 ensures robust RA signaling required for SHF pattern and induction of endodermal *shh/dhh* expression, and second, by cooperating with Hh to activate mesodermal expression of *Wnt2/2b* which promotes pulmonary induction.

## Tbx5 directly activates *Aldh1a2* transcription and indirectly represses *Fgf8* via RA

In preliminary experiments, we found that expression of a doxycycline (Dox) inducible *Tbx5* transgene during the directed differentiation of mouse embryonic stem cells (mESCs) into cardiac fate (*Kattman et al., 2011*; *Steimle et al., 2018*) was sufficient to increase *Aldh1a2* expression and suppress *Fgf8* and *Fgf10* levels (*Figure 4A*). However, in these experiments, it was unclear whether Tbx5 regulated *Aldh1a2* or *Fgf* expression directly or indirectly.

We therefore examined whether Tbx5 was sufficient to directly activate *aldh1a2* transcription in *Xenopus*. We injected RNA encoding a dexamethasone (DEX) inducible Glucocorticoid receptor (GR)-Tbx5 fusion protein (*Horb and Thomsen, 1999*) into either the anterior or posterior mesoderm. We then induced GR-Tbx5 nuclear translocation at gastrula stage before endogenous *tbx5* is normally expressed by addition of DEX, with or without the translation inhibitor cycloheximide (CHX) to block secondary protein synthesis (*Figure 4—figure supplement 1A*). GR-Tbx5 activated precocious *aldh1a2* transcription in both the anterior and posterior tissue even in the presence of CHX, demonstrating direct activation (*Figure 4B*). In-situ hybridization of NF34 embryos confirmed robust, ectopic activation of *aldh1a2* by GR-Tbx5 (*Figure 4C*). In contrast, suppression of *fgf8* transcription by GR-Tbx5 was sensitive to CHX, demonstrating indirect repression (*Figure 4—figure supplement 1A,B*). We hypothesized that Tbx5 indirectly represses *fgf8* via Aldh1a2-dependent RA production since RA is known to directly repress *Fgf8* transcription in the mouse SHF (*Kumar et al., 2016*). We tested this by inhibiting Aldh activity with DEAB which prevented the suppression of *fgf8* by GR-Tbx5 (*Figure 4—figure supplement 1A-C*). These data demonstrate that Tbx5 directly activates *aldh1a2* transcription and indirectly suppresses *fgf8* expression via RA.

## Tbx5 maintains *Aldh1a2* transcription via an evolutionarily conserved intronic enhancer

We next sought to identify *Aldh1a2* enhancers that are directly regulated by Tbx5, predicting that these would be evolutionarily conserved across terrestrial vertebrates. Since a number of putative enhancers have been documented for the mouse *Aldh1a2* locus (*Castillo et al., 2010*; *Vitobello et al., 2011*; *Huang et al., 2012*), we focused on the murine genome. To identify Tbx5-bound enhancers in the CP lineage, we performed Tbx5 chromatin immunoprecipitation followed by high-throughput sequencing (ChIP-seq) of E14.5 fetal mouse lungs as lung mesenchyme is derived from the E9.5 CPPs (*Peng et al., 2013*). ChIP-seq uncovered five Tbx5-bound regions at the *Aldh1a2* locus. Comparing the lung ChIP-seq data to our previously published Tbx5 ChIP-seq from E14.5 heart (*Steimle et al., 2018*), we found that four of the five Tbx5-bound sites were lung-specific and not bound by Tbx5

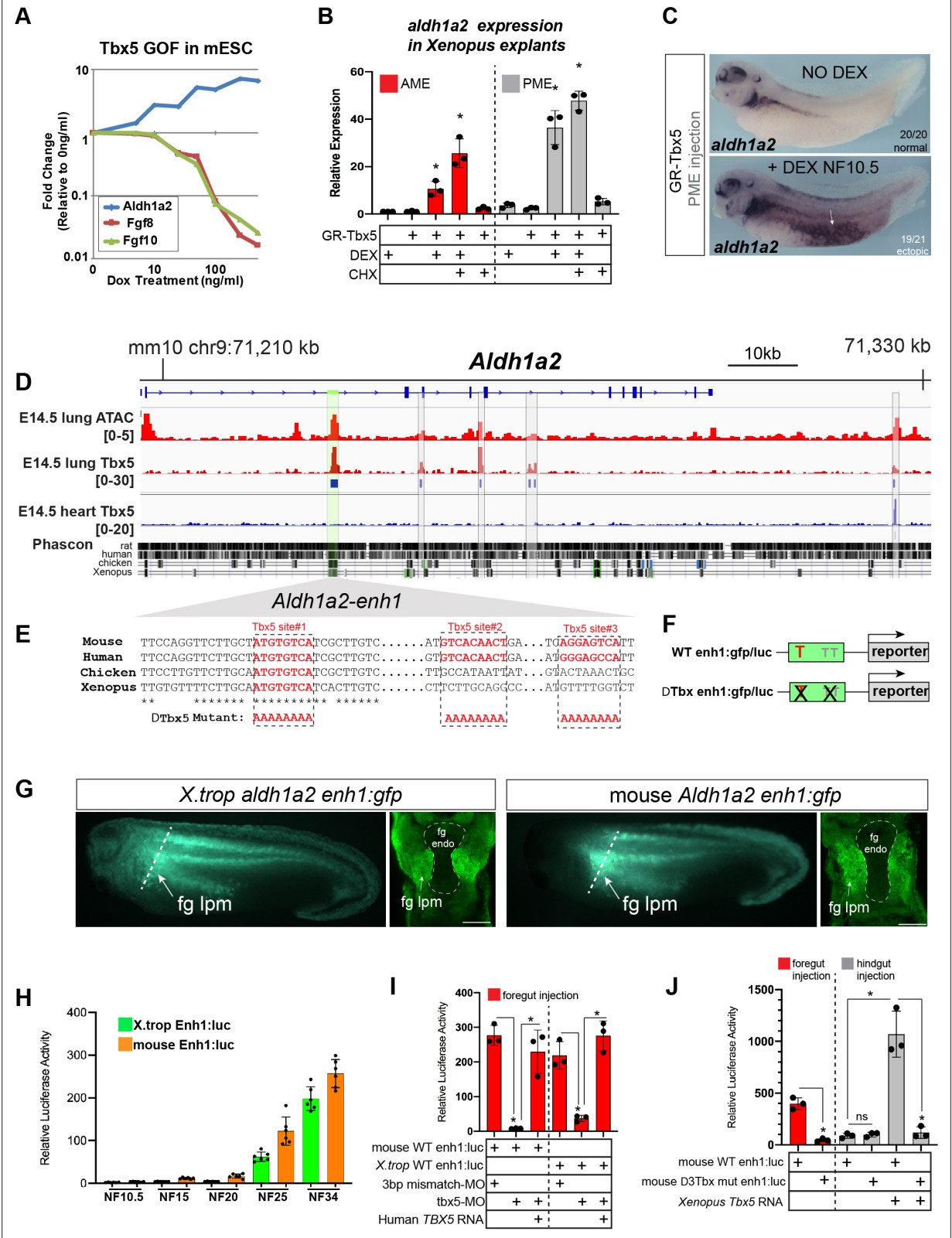

**Figure 4.** Tbx5 directly activates *aldh1a2* transcription via an evolutionarily conserved first intron enhancer. (**A**) Dox-inducible Tbx5 activated *Aldh1a2* expression and repressed *Fgf8* and *Fgf10* during the directed differentiation *of* mouse embryonic stem cells (mESCs) into cardiac progenitors in a dose-dependent manner. (**B**) Tbx5 directly activated *aldh1a2* expression in *Xenopus* anterior or posterior mesendoderm (AME, PME) explants. RT-qPCR shows that *aldh1a2* transcription was induced by DEX activated GR-Tbx5 in the presence of translation inhibitor CHX. Graphs show mean relative expression ±

*Figure 4 continued on next page*

Figure 4 continued

standard deviation from N=3 biological replicates, four explants/replicate. *p<0.05, pair-wise Student's t-test relative to uninjected, untreated explants. (**C**) Whole-mount in-situ hybridization of *aldh1a2* expression of *X. laevis* NF34 embryos injected with GR-Tbx5 (100 pg) into the PME with or without DEX. (**D**) Genome browser of the mouse *Aldh1a2* locus showing Tbx5 ChIP-seq tracks from E14.5 mouse lung (GSE167207) and E14.5 mouse heart (**Burnicka-Turek et al., 2020**, GSE139803) as well as ATAC-seq track from the ENCODE project (**Castillo et al., 2010**; **Davis et al., 2018**; ENCSR335VJW). Tbx5 ChIP-seq peaks in the E14.5 lung are indicated in blue. Phascon multiple species conservation track shows that the prominent Tbx5-bound first intron enhancer (enh1) is evolutionarily conserved from mammals to *Xenopus*. (**E**) Multiple species sequence alignment of enh1 reveals one Tbx5 DNA-binding site conserved from mammals to *Xenopus* and two additional mammalian-specific Tbx5 sites, which were mutated in reporter constructs. (**F**) Schematic of the Wild-type (WT) and mutant (ΔTbx) enh1:gfp and enh1:luciferase reporter constructs. (**G**) Both the *Xenopus* and mouse intronic enh1 enhancer are sufficient to drive GFP expression in the foregut lpm in *Xenopus* transgenic assays. (**H**) Time course of *Xenopus* and mouse enh1:luc reporter activity injected into *X. laevis* CP-foregut tissue, reflects endogenous Tbx5-dependent *aldh1a2* expression between NF25 and NF34. Graphs show mean relative luciferase activity ± standard deviation. N=5 biological replicates/time point with five embryos/replicate. *p<0.05, parametric two-tailed paired t-test. (**I**) The *Xenopus* and mouse *aldh1a2 enh1* reporter constructs are regulated by Tbx5. Graphs show relative mean luciferase activity ± standard deviation of reporters injected into CP-foregut tissue with control mm-MO, Tbx5-MO, and/or human *TBX5* RNA. N=3 biological replicates/time point with five embryos/replicate. *p<0.05, parametric two-tailed paired t-test. (**J**) The three putative Tbx5 motifs in the mouse *aldh1a2-enh1* enhancer are required for reporter activity in the CP-foregut tissue and Tbx5-dependent activation in the hindgut. Graphs show mean relative luciferase activity ± standard deviation. N=5 biological replicates/time point with five embryos/replicate. *p<0.05, parametric two-tailed paired t-test. Also see *Figure 4—figure supplement 1*, *Figure 4—figure supplement 2* and related source data files. CP, cardiopulmonary.

The online version of this article includes the following figure supplement(s) for figure 4:

**Source data 1.** *Figure 4B*.

**Source data 2.** *Figure 4H* Luciferase source data.

**Source data 3.** *Figure 4I*.

**Source data 4.** *Figure 4J*.

**Figure supplement 1.** GR-Tbx5 indirectly suppresses *fg8* in *Xenopus* via RA and additional analysis of the first intron Tbx5 enh1 enhancer.

**Figure supplement 1—source data 1.** RT-qPCR source data of *Xenopus* anterior (AME) or posterior (PME) NF10.5 mesendoderm explants injected with GR-Tbx5 RNA and treated±DEX, CHX, or DEAB.

**Figure supplement 1—source data 2.** Luciferase source data of mouse and *Xenopus tropicalis* enh1 reporter activity in foregut and hindgut at NF25, NF34.

**Figure supplement 2.** Multiple species sequence alignment of the *Aldh1a2* enh1 enhancers.

in the fetal heart (*Figure 4D*). Among the four Tbx5-bound sites, only one peak in the *Aldh1a2* first intron, which we refer to as enh1 (for 'enhancer 1;' *Figure 4D*, *Figure 4—figure supplement 2*), showed strong evolutionarily conservation from mammals to *Xenopus*. The enh1 region also had a strong ATAC-seq peak in E14.5 lungs consistent with open enhancer chromatin (*Figure 4D*). Sequence analysis of enh1 revealed multiple predicted Tbx5 DNA-binding motifs, one of which was perfectly conserved amongst human, mouse, chicken, and *Xenopus* (*Figure 4E*; *Figure 4—figure supplement 2*).

We tested the ability of both the mouse and *X. trop* enh1 intronic enhancers to drive reporter expression in *Xenopus* transgenics and luciferase reporter assays (*Figure 4F–J*). In transgenics, both the mouse and *X. trop* enh1 enhancers drove GFP expression in the foregut lpm/pSHF at NF34 overlapping endogenous *aldh1a2* (*Figure 4G*; *Figure 4—figure supplement 1D-E''*). We noted however that the transgene expression domain in the lpm was broader than endogenous *aldh1a2* (*Figure 4G*), moreover, enh1 did not drive expression in the somites (*Figure 4—figure supplement 1D-E*), suggesting other enhancers must refine the lpm expression and promote somitic expression in vivo. To quantitatively assess temporal and spatial enhancer activity, we micro-injected enh1 reporters into blastomeres targeting the future CP-foregut or hindgut regions and assayed luciferase activity at a range of developmental stages, from gastrula to tailbud (*Figure 4H*; *Figure 4—figure supplement 1F,G*). Neither the mouse nor *X. trop* enh1 enhancers drove significant reporter activity during early development at NF10.5, NF15, or NF20; however, at NF25 and N34 both the mouse and *X. trop* enh1 enhancers were active in the foregut but not hindgut (*Figure 4H*; *Figure 4—figure supplement 1F,G*). This coincides with the timing at which endogenous *aldh1a2* expression is Tbx5-dependent (*Figure 2*). Taken together, these data demonstrate that the evolutionarily conserved enh1 regulates the temporal and spatial transcription of *aldh1a2* in the foregut lpm/pSHF.

We next tested Tbx5 regulation of the enh1 enhancer by combining reporter assays with LOF or gain-of-function (GOF) experiments. Tbx5-MO knockdown resulted in a dramatic reduction of the

mouse and *X. trop* enh1 reporter activity in CP-foregut tissue at NF34, which was rescued by injection of human *TBX5* RNA (*Figure 4I*). Moreover, injection of *Xenopus* or human *TBX5* RNAs were sufficient to ectopically induce robust enh1 reporter activity in hindgut tissue, which does not express endogenous *tbx5* (*Figure 4J*). Mutation of the single Tbx5-binding site that was perfectly conserved amongst human, mouse, chicken, and *Xenopus* enh1 resulted in a 48% (p=0.0046) and 60% reduction (p=0.0031) of the mouse and frog reporter activity in the foregut respectively, and also significantly blunted their response to ectopic Tbx5 in the hindgut (*Figure 4—figure supplement 1H,I*). Mutation of all three putative Tbx5 motifs conserved amongst mammals (*Figure 4J*) largely abolished reporter activity in both the foregut and hindgut (*Figure 4J*). We conclude Tbx5 directly maintains *Aldh1a2* expression via multiple T-box motifs found in an evolutionarily conserved first intron enhancer.

## FGF gain-of-function phenocopies Tbx5-loss-of-function in *Xenopus*

In light of the finding that Tbx5-dependent RA signaling suppresses *fgf8* and *fgf10*, we tested if a temporal FGF GOF would phenocopy Tbx5 LOF (*Figure 5A*). We treated WT CP-foregut explants with recombinant FGF8 protein from NF20 to NF34, the period when exogenous RA was sufficient to rescue Tbx5 LOF. As predicted FGF8 treatment largely phenocopied Tbx5 depletion with increased expression of aSHF/pharyngeal markers *tbx1, fgf10, spry2,* and *cyp26a1, as well as* reduced expression of pSHF and pulmonary endoderm genes *wnt2b, shh, gli1,* and *nkx2-1* (*Figure 5B*). We also observed reduced expression of *tbx5* and *aldh1a2* consistent with a feedback loop where FGF restricts Tbx5/Aldh1a2-mediate RA signaling (*Figure 5C*).

FGF signaling is known to promote the expression of RA-degrading Cyp26 enzymes (*Shiotsugu et al., 2004*; *Deimling and Drysdale, 2011*; *Rydeen and Waxman, 2016*), but it is unclear whether this is by direct transcriptional regulation. Therefore, we repeated the FGF8 experiments in the presence of CHX and found that indeed *cyp26a1* and *cyp26c1* were still upregulated by FGF8, demonstrating direct activation (*Figure 5D*). In contrast, the ability of FGF8 to suppress *shh* was CHX sensitive, demonstrating indirect repression (*Figure 5D*). We hypothesized that FGF8 indirectly suppresses expression of *shh* and other RA-dependent pSHF genes by promoting Cyp26-mediated RA degradation (*Figure 5C*). To test this, we treated CP-foregut explants with both FGF8 and the CYP enzyme inhibitor ketoconazole (keto). Keto blocked the ability of FGF8 to suppress *shh, dhh, tbx5, aldh1a2, wnt2b,* and *nkx2-1* (*Figure 5B*), indicating that FGF indeed acts via Cyp-dependent RA degradation. Consistent with Cyp-mediated RA degradation being a major factor in endogenous CP patterning, keto treatment alone elevated expression of pSHF (*tbx5, aldh1a2,* and *wnt2b*) and pulmonary endoderm genes (*shh* and *nkx2-1*), whilst decreasing aSHF markers (*fgf8, fgf10,* and *tbx1*) (*Figure 5B*), similar to exogenous RA treatment (*Figure 3*). Moreover, knockdown of Cyp26a1 and Cyp26c1 by targeted MO injection phenocopied the ketoconazole treatment (*Figure 6—figure supplement 1A,B*). Interestingly, inhibition or knockdown of Cyp26 resulted in increased *tbx5* levels suggesting that RA promotes its expression. Indeed, in-situ hybridization showed *tbx5* expression in the pSHF/foregut lpm, but not in the FHF/heart tube, reduced by DEAB indicating that it requires RA (*Figure 5E*). Combined with our finding that Tbx5 directly maintains *aldh1a2* expression, these data identify a RA-Tbx5 positive feedback loop in the pSHF.

## RA directly promotes *shh* transcription through the evolutionarily conserved MACS1 endoderm enhancer

Our data suggest that RA from the Aldh1a2-expressing lpm is a likely candidate to activate Hh ligand expression in the endoderm. We tested whether exogenous RA could directly activate *shh* and *dhh* transcription in *Xenopus* foregut endoderm explants where the *tbx5/aldh1a2*+ lpm, the source of endogenous RA, had been removed (*Figure 6A and B*). Without the RA-producing lpm, the foregut endoderm did not express *shh* nor *dhh*; however, addition of exogenous RA rescued their expression, even in the presence of CHX, demonstrating direct activation (*Figure 6B*). As controls, RA also rescued expression of the known direct RA-target *hnf1b*, whereas the known indirect target *ptf1a* was not rescued in the presence of CHX (*Figure 6B*).

Previous work has identified an evolutionarily conserved distal *Shh* enhancer called MACS1 (for **m**ammalian-**a**mphibian-**c**onserved **s**equence **1),** which is located more than 800 kb from *Shh*, within an intron of the *Rnf32* gene (*Sagai et al., 2009*; *Tsukiji et al., 2014*; *Sagai et al., 2017*). The MACS1 enhancer is able to dive transcription in mouse foregut endoderm but the signals and TFs that control

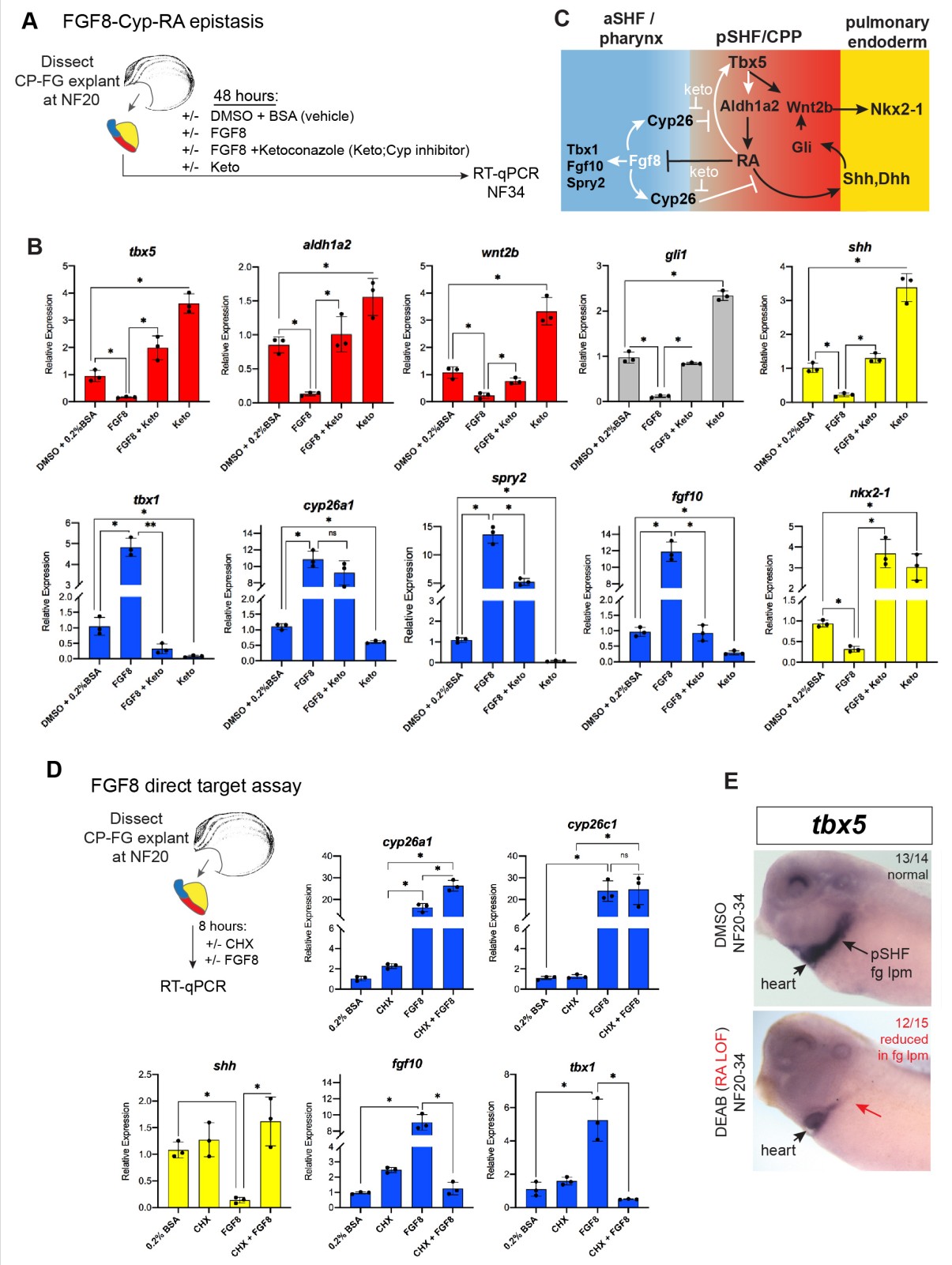

**Figure 5.** FGF8 gain-of-function (GOF) phenocopies Tbx5 loss-of-function in *Xenopus*. (**A**) Schematic of FGF8 GOF assay in *Xenopus* cardiopulmonary foregut (CP-FG) explants dissected at NF20, and treated with vehicle controls (DMSO +0.2% BSA) or the indicated combinations of 100 ng/ml FGF8b and/or 0.5 µM ketoconazole (Cyp-inhibitor), harvested at NF34, and analyzed via RT-qPCR. (**B**) RT-qPCR showing mean relative expression of genes for pSHF (red), aSHF (blue), and pulmonary endoderm (yellow), ± standard deviation from N=3 biological replicates (four explants/replicate).

*Figure 5 continued on next page*

*Figure 5 continued*

*p<0.05, parametric two-tailed paired t-test. (**C**) Model depicting the observed FGF8 GOF results. White arrows indicate relationships tested in these experiments. (**D**) FGF8 direct target gene assay in *Xenopus* CP foregut explants, demonstrating that FGF8 directly activates *cyp26a1, cyp26c1* and indirectly suppresses *shh*. Explants dissected at NF20 were pre-treated with 1 µM cycloheximide (CHX) for 2 hr prior to culture in 100 ng/ml FGF8b+CHX for 6 hr followed by RT-qPCR analysis. Graphs display mean relative expression ± standard deviation from N=3 biological replicates that contained four explants/replicate. *p<0.05, parametric two-tailed paired t-test. (**E**) RA signaling is required for the *tbx5* expression in the fg lpm/pSHF domain, but not the heart. Embryos were cultured in 10 µM DEAB from NF20 to NF34 and assayed by in-situ hybridization. Number of embryos assayed and with the observed expression pattern is indicated. Also see ***Figure 5—figure supplement 1*** and related source data files. aSHF, anterior second heart field; pSHF, posterior second heart field.

The online version of this article includes the following figure supplement(s) for figure 5:

**Source data 1.** *Figure 5B*.

**Source data 2.** *Figure 5D*.

**Figure supplement 1.** Cyp26a1/c1 morpholino (MO) knockdown phenocopies Cyp inhibitor treatment.

**Figure supplement 2—source data 1.** RT-qPCR source data of FGF8 treated *Xenopus* CP-foregut explants±cyp26 a1/c1 MO.

*Shh* expression via the MACS1 enhancer are unknown. An analysis of publicly available ChIP-seq data from human foregut endoderm, differentiated from pluripotent stem cells (hPSCs) in part by RA treatment (***Vinckier et al., 2020***; ***Wang et al., 2015***), revealed binding of the RA nuclear receptor RXR at the human *SHH* MACS1 enhancer as well as H3K4me1 and H3K27ac1, epigenetic marks indicative of enhancer activation (***Figure 6C***). Sequence analysis of the MACS1 enhancer predicted multiple RXR/RAR nuclear RA receptor half sites (***Penvose et al., 2019***), two of which were evolutionarily conserved between human, mouse, chicken, and *Xenopus* (***Figure 6D***; ***Figure 6—figure supplement 1***), suggesting that RA directly activates *SHH* transcription.

We functionally interrogated human and *X. tropicalis SHH* MACS1 enhancer activity in *Xenopus* luciferase assays (***Figure 6E***) and found both could drive robust reporter activity in foregut but not hindgut endoderm, demonstrating spatial specificity (***Figure 6E***). Disruption of endogenous RA signaling via DEAB treatment (NF20–34) or injection of dominant-negative RAR alpha RNA (dN-RARa) abolished human and *X. trop* MACS1 enhancer activity (***Figure 6E***). Moreover, exogenous RA could activate the enhancer in foregut explants lacking the RA-producing lpm. Mutation of the two highly conserved RAR/RXR half sites in the MACS1 enhancers dramatically reduced reporter activity in the foregut (***Figure 6E***) as well as a significantly blunted activation by exogenous RA in isolated endoderm explants (***Figure 6—figure supplement 2***). These data demonstrate that RA signaling directly stimulated *shh* transcription in foregut endoderm, via conserved RAR/RXR motifs in the *shh* MACS1 enhancer.

## Discussion

### Tbx5 regulates a RA-HH-Wnt GRN that coordinates SHF patterning and pulmonary specification

Our findings reveal complex and evolutionarily conserved interconnected signaling networks downstream of Tbx5 that coordinate early development of the cardiac and pulmonary systems (modeled in ***Figure 7***). We identify the following aspects of this SHF mesoderm—pulmonary endoderm signaling network: (1) Direct Tbx5 activation of an *aldh1a2* enhancer, which maintains *aldh1a2* transcription in pSHF mesoderm; RA is in turn required to maintain *tbx5* expression in the pSHF, establishing a positive feedback loop between Tbx5 and RA; (2) Tbx5-RA and FGF-Cyp form mutually antagonistic modules, with the Tbx5-RA loop promoting pSHF/CPP identity and suppressing aSHF fate, and Cyp-mediated RA degradation refining the spatial domain of RA activity; and (3) Direct RXR/RAR activation of the MACS1 enhancer at the shh locus, which provides a mechanism underlying the cell-non-autonomous activation of endodermal *Hh* ligand expression by Tbx5/Aldh1a2-dependent RA signaling in the pSHF. Reception of Hh signaling in the pSHF mesoderm activates Gli TFs, which cooperate with Tbx5 to directly activate *wnt2/2b* transcription; Wnt2/2b then induce pulmonary fate in the foregut endoderm (***Hoffmann et al., 2014***; ***Rankin et al., 2016***; ***Steimle et al., 2018***; ***Goddeeris et al., 2008***). Thus, during CP development, Tbx5 regulates the production of three key paracrine signals, RA and Wnt directly

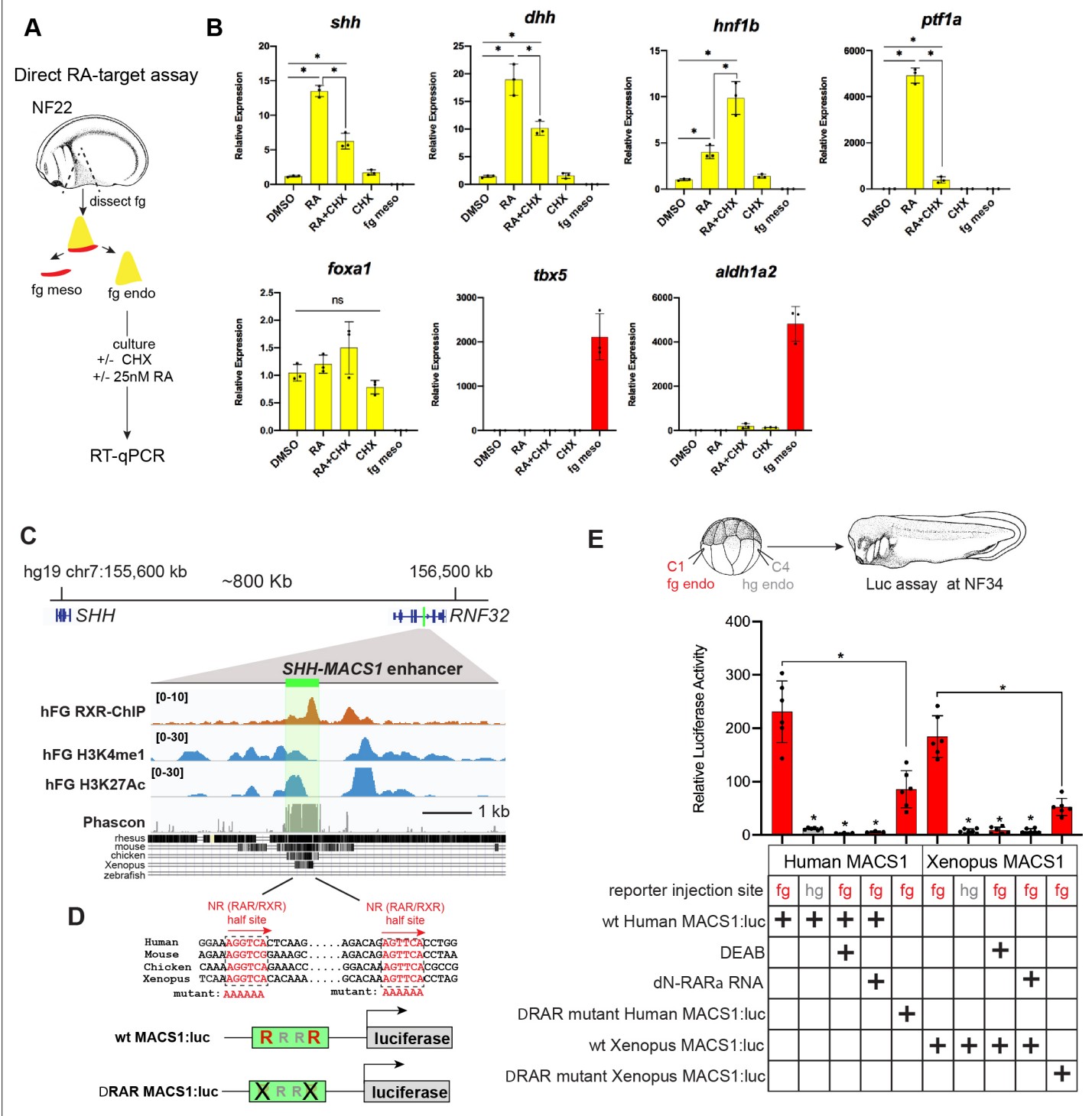

**Figure 6.** RA-RAR directly activates *shh* transcription in the *Xenopus* foregut endoderm via an evolutionarily conserved MACS1 enhancer. (**A**) Schematic of direct RA target gene assay. Foregut endoderm (fg endo; yellow) was dissected from foregut mesoderm (fg meso; red) at NF25, pre-treated with 1 μM cycloheximide (CHX) for 2 hr prior to culture in 25 nM RA + CHX (or DMSO vehicle control) for 6 hr followed by RT-qPCR analysis. (**B**) RA directly activates *shh* and *dhh* expression in the presence of CHX. Graphs show mean relative expression ± standard deviation from N=3 biological replicates (four explants/replicate). Endoderm genes are shown in yellow, mesoderm makers in red confirm dissections. *p<0.05, parametric two-tailed paired t-test. (**C**) Genome browser of the human *SHH* locus showing the evolutionarily conserved MACS1 distal enhancer (green shading) embedded in an intron of the *RNF32*. Published ChIP-seq tracks of RXR, H3K4me1, and H3K27ac1 from hPSC-derived foregut endoderm (*Vinckier et al., 2020*, GSE104840; *Wang et al., 2015*, GSE54471). (**D**) MACS1 enhancer contains multiple RAR/RXR DNA-binding half sites, two of which are highly conserved. Schematics of the wild-type and mutant MACS1:luciferase reporter constructs. (**E**) Luciferase reporter assay in *Xenopus* show that the Human and *X*.

*Figure 6 continued on next page*

*Figure 6 continued*

*tropicalis* MACS1 enhancers are activated by RA via the RAR/RXR DNA-binding sites. 50 pg of MACS1:luciferase reporter +5 pg pRL-TK reporter were microinjected±250 pg of dominant-negative RARa RNA into either the C1 foregut (fg; red bars) or C4 hindgut (hg; gray bars) blastomeres and luciferase activity was assayed at NF34. 10 µM DEAB treatment was from NF20 to NF34. Mean relative luciferase activity ± standard deviation, from N=6 biological replicates/time point with five embryos/replicate. *p<0.05, parametric two-tailed paired t-test relative to WT MACS1:luc in the foregut (fg). Also see *Figure 6—figure supplement 1*, *Figure 6—figure supplement 2* and related source data files. ns, not significant.

The online version of this article includes the following figure supplement(s) for figure 6:

**Source data 1.** *Figure 6B*.

**Source data 2.** *Figure 6E*.

**Figure supplement 1.** Multiple species alignment of the *Shh* MACS1 enhancers.

**Figure supplement 2.** Conserved RAR/RXR sites in the *Shh* MACS1 enhancer are required for RA-mediated activation.

**Figure supplement 2—source data 1.** Luciferase source data of wild-type and mutant Shh-MACS1 reporters in *Xenopus* foregut endoderm explants ±25 nM RA.

via Tbx5-dependent enhancers and Hh indirectly via a RA-dependent enhancer. These interdependent signaling loops ensure that the lung primordia and pSHF-derived atria and pulmonary vessels from adjacent to one another, in preparation for the coordinated morphogenesis and functional integration of these two organ systems during development.

## T-box TFs and RA: a conserved regulatory node disrupted in cardiopulmonary and limb birth defects

Integrated regulatory loops between Tbx5, RA, and FGF regulate limb development and lung branching morphogenesis in addition to SHF cardiac development (*Nishimoto et al., 2015*; *Arora et al., 2012*; *Feneck and Logan, 2020*). We show that Tbx5 and RA form a positive feedforward loop in the pSHF; in this domain, Tbx5 directly maintains Aldh1a2-dependent RA production while RA maintains *tbx5* expression. This is consistent with reports that RA is required for the expression of Tbx5 in SHF but not the FHF during early in mouse heart development (*De Bono et al., 2018*; *Stefanovic et al., 2020*). We predict that this is equivalent to the RA-dependent maintenance of *tbx5* that we observed in *Xenopus*. In the developing limb bud, RA response elements in a regulatory element at *Tbx5* are required for enhancer activity and other enhancers at the *Tbx5* locus have been identified that can activate transcription in the heart and

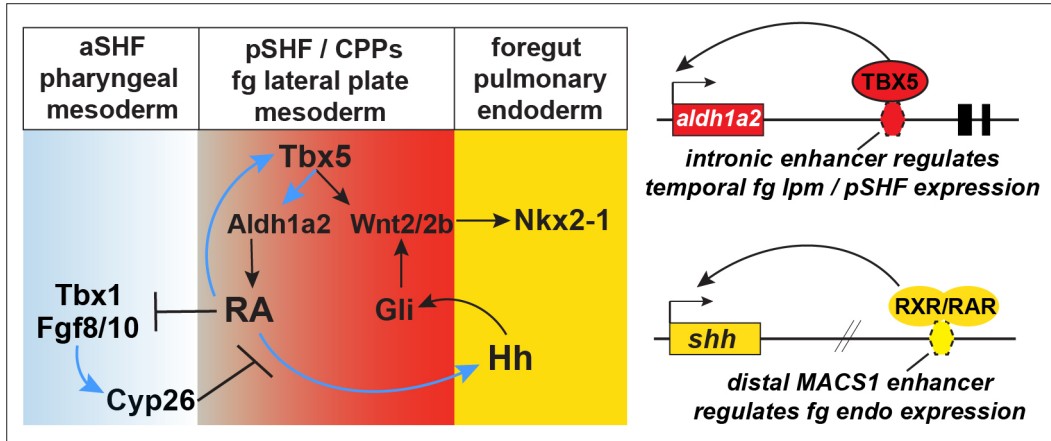

**Figure 7.** Model of the Tbx5 regulated GRN coordinating SHF pattering and pulmonary induction. Our data indicate that between NF25and NF35 in *Xenopus* and around E9.5 in mice, Tbx5 directly maintains *Aldh1a2* expression and a RA-Tbx5 positive feedback loop in the pSHF, which is necessary for *Hh* ligand expression, Wnt2/2b-dependent pulmonary fate induction, and SHF patterning. Blue arrows in the model indicate relationships demonstrated in this study. Tbx5/Aldh1a2-dependent RA signaling restricts FGF/Cyp activity in the aSHF, promotes pSHF identity, and drives expression of *shh* in pulmonary foregut endoderm. The *aldh1a2* enh1 enhancer is directly regulated by Tbx5 and the *shh* MACS1 enhancer is regulated by RA/RXR/RAR. aSHF, anterior second heart field; GRN, gene regulatory network; pSHF, posterior second heart field.

limb (*Minguillon et al., 2012*; *Smemo et al., 2012*; *Cunningham et al., 2018*), it remains to be determined whether these enhancers are also directly regulated by RA/RAR/RXR or control expression in the pSHF. Regardless, Tbx5 and RA are from a shared module in both SHF and limb development (*Nishimoto et al., 2015*). Limb defects and atrioventricular septal defects, caused by altered pSHF development, are both a facet of the phenotypic spectrum observed in Holt–Oram syndrome in human patients with *TBX5* mutations (*Steimle and Moskowitz, 2017*). This raises the intriguing possibility that Tbx5-RA interactions were an evolutionary innovation in both limb and CP mesoderm in the adaptation to terrestrial life and that disrupting the Tbx5-RA feedforward loop is a component of *TBX5*-associated birth defects. Overall, this work provides a framework for understanding the developmental basis of the human birth defects observed in Holt–Oram Syndrome.

## Enhancers controlling the reciprocal RA-HH mesoderm and endoderm signaling

We found that Tbx5-dependent RA signaling directly promotes *Shh* expression in foregut endoderm via RAR/RXR half-site motifs in the evolutionarily conserved *Shh* MACS1 enhancer (*Sagai et al., 2009*; *Tsukiji et al., 2014*; *Sagai et al., 2017*; *Penvose et al., 2019*). Other endodermal TFs that contribute to *Shh* expression in the foregut have been described, including FoxA2, Meis3, Islet1, Tbx2, and Tbx3 (*Lin et al., 2006*; *diIorio et al., 2007*; *Tamplin et al., 2008*; *Mesbah et al., 2012*; *Lee et al., 2019*; *Genga et al., 2019*), but whether they directly function on the *Shh* MACS1 enhancer, potentially in cooperation with RA/RAR/RXRs, remains unknown.

Previous studies have defined multiple enhancers that regulate different temporal and spatial expression domains of *Aldh1a2* during development, with input from T-box TFs as a reoccurring theme. For example, the T-box TFs VegT, Eomesodermin, and Brachyury, regulate *aldh1a2* in the *Xenopus* gastrula mesoderm via cis-regulatory elements near the promoter (*Gentsch et al., 2013*; *Faial et al., 2015*; *Tosic et al., 2019*). Subsequent expression of *Aldh1a2* in the paraxial mesoderm and early lpm is promoted by Hox/Pbx/Meis TF complexes acting on a first intron enhancer (*Vitobello et al., 2011*) that is distinct from the enh1 enhancer identified here. Indeed, the Tbx5-responsive enh1 transgene did not drive expression in the paraxial mesoderm-derived somites.

The fidelity of these T-box/RA modules is essential for avoiding common cardiovascular birth defects affecting both the aSHF and pSHF. While Tbx5 and RA act in a positive feedback loop during pSHF patterning, Tbx1 and RA have an antagonistic relationship in the aSHF. Interestingly Tbx1, which can act as a transcriptional repressor, is known to spatially restrict *Aldh1a2* expression (*Guris et al., 2006*; *Aggarwal et al., 2006*; *Ryckebüsch et al., 2010*), although it is currently unknown whether this activity is direct. On the other hand, RA suppresses *tbx1* expression in both *Xenopus* pSHF (our study) and in mice (*Ryckebüsch et al., 2010*). In mouse, both loss of *Tbx1* and aberrant RA synthesis can result in cardiovascular defects similar to human DiGeorge syndrome patients. Moreover, genetically removing one copy of *Aldh1a2*, thereby reducing the level of RA, ameliorates the cardiovascular malformations in Tbx1 heterozygous embryos (*Ryckebüsch et al., 2010*; *Vermot et al., 2003*). Taken together, these observations suggest that the opposing actions of Tbx5 and Tbx1 act as a mechanistic toggle, wherein RA is activated by Tbx5 to promote the CP program in the pSHF and restrict aSHF identity. In the aSHF Tbx1 suppresses RA production and the CP program. We speculate that Tbx1 and Tbx5 may engage T-box elements in the same enh1 enhancer, with Tbx5 promoting *Aldh1a2* in the pSHF and Tbx1 inhibiting *Aldh1a2* transcription in the aSHF as a transcriptional mechanism contributing to posterior/anterior patterning of the SHF.

Identification of the specific transcriptional enhancers that mediate the reinforcing signaling loops that pattern the SHF is essential for the genotype-phenotype interpretation of animal model and patient CP defects. The enhancers we identified by which Tbx5 directly activates *aldh1a2* transcription and RA directly activates *shh* are both highly conserved amongst air breathing terrestrial vertebrate species, suggesting a potential role in CP evolution. Previous work has identified a single nucleotide variant in a TBX5 enhancer that contributes to human CHD (*Smemo et al., 2012*). Identification of the enhancers modulating the essential signaling pathways for heart development will contribute to the curation of whole-genome sequencing, refining the search space for functional non-coding variants

and allowing the nomination of non-coding SNPs that may alter the function of known enhancers and thereby contribute to CHD risk.

# Materials and methods

## Key resources table

| Reagent type (species) or resource | Designation | Source or reference | Identifiers | Additional information |
|---|---|---|---|---|
| Strain, strain background (*Xenopus tropicalis*, females) | Wild-type adult females | Nasco | LM00823 | |
| Strain, strain background (*X. tropicalis*, males) | Wild-type adult males | Nasco | LM00822 | |
| Strain, strain background (*X. laevis*, females) | Wild-type adult females | Nasco | LM00531 | |
| Strain, strain background (*X. laevis*, males) | Wilt-type adult males | Nasco | LM00715 | |
| Genetic reagent (*X. tropicalis*) | *Xtr.Tg(WntREs: dEGFP)*[Vlemx] | National *Xenopus* Resource (NXR) Center, Woods Hole, MA | RRID:NXR_1094 | *X. tropicalis* Wnt/Bcat reporter line |
| Genetic reagent (*X. laevis*) | *Xla.Tg(WntREs: dEGFP)*[Vlemx], | NXR | RRID:NXR_0064 | *X. laevis* Wnt/Bcat reporter line |
| Genetic reagent (*X. laevis*) | *Xla.Tg.(nkx2-5:GFP)*[Mohun] | NXR | RRID:NXR_0030 | *X. laevis* Nkx2-5:GFP reporter line |
| Strain, strain background (*Mus musculus*) | CD-1 | Charles River Labs | Strain Code 022 **RRID:IMSR_CRL:022** | WT mice |
| Genetic reagent (*M. musculus*) | Shh[tm1(EGFP/cre)Cjt] | Jax Labs | JAX: 005622 **RRID:IMSR_JAX:005622** | *Shh:GFP* mice |
| Cell line (*M. musculus*) | *Tbx5*OE-mESC line | **Steimle et al., 2018** | **Steimle et al., 2018** | |
| Chemical compound, drug | Doxycycline | Sigma-Aldrich | D9891 | |
| Antibody | (Rabbit polyclonal) anti-Aldh1a2 | Abcam | ab96060, RRID: AB_10679336 | IF (1:500) |
| Antibody | (Mouse Monoclonal) Anti-Aldh1/2 | Santa Cruz Biotechnology | sc-166362, RRID: AB_2009458 | IF (1:500) |
| Antibody | (Mouse monoclonal) anti-Sox2 | Abcam | ab79351; RRID: AB_10710406 | IF (1:1000) |
| Antibody | (Rabbit polyclonal) anti-Nkx2-1 (H-190) | Santa Cruz Biotechnology | sc-13040X; RRID: AB_793532 | IF (1:500) |
| Antibody | (Mouse monoclonal) anti-Fibronectin (4H2) | Developmental Studies Hybridoma Bank | DSHB #4H2; RRID: AB_2721949 | IF (1:2000) |
| Antibody | (Chicken polyclonal) anti-GFP | Aves Labs | GFP-1020; RRID: AB_10000240 | IF (1:1000) |
| Antibody | (Goat polyclonal) anti-Tbx5 | Santa Cruz Biotechnology | sc-17866, RRID: AB_2200827 | IF (1:300) ChIP: 5 µg |
| Recombinant DNA reagent | pI-SceI-d2EGFP plasmid | Addgene | Addgene_32674 | For meganuclease transgenics |

*Continued on next page*

*Continued*

| Reagent type (species) or resource | Designation | Source or reference | Identifiers | Additional information |
|---|---|---|---|---|
| Recombinant DNA reagent | pRL-TK (plasmid) | Promega | E2241 | |
| Recombinant DNA reagent | pGL4.23 luc2/miniP (plasmid) | Promega | E8411 | |
| Recombinant DNA reagent | pCS2+ GR-xTbx5 | Addgene | Addgene 117248 | |
| Recombinant DNA reagent | pCS2+ xTbx5 | Addgene | Addgene 117247 | |
| Recombinant DNA reagent | pCSf107mT-Gateway-3'myc | Addgene | Addgene 67617 | |
| Recombinant DNA reagent | pENTR223 Human TBX5 | Horizon Discovery | OHS5894-202500411 | |
| Commercial assay or kit | Gateway LR Clonase II enzyme mix | Thermo Fisher Scientific | 11791020 | |
| Commercial assay or kit | mMessage mMachine SP6 RNA synthesis kit | Thermo Fisher Scientific | AM1340 | |
| Peptide, recombinant protein | FGF8b | R&D Systems | 423-F8-025 | |
| Peptide, recombinant protein | WNT2B | R&D Systems | 3900-WN-025 | |
| Commercial assay or kit | TRIzol | Thermo Fisher Scientific | 15596018 | |
| Commercial assay or kit | Direct-zol Miniprep plus kit | Thermo Fisher Scientific | R2070 | |
| Commercial assay or kit | Superscript VILO mastermix | Thermo Fisher Scientific | 11755050 | |
| Commercial assay or kit | PowerUP 2× SYBR Green MasterMix | Thermo Fisher Scientific | A25742 | |
| Commercial assay or kit | Firefly Luciferase 2.0 kit | Biotium | 30085-1 | |
| Commercial assay or kit | Renilla Luciferase 2.0 kit | Biotium | 30082-1 | |
| Chemical compound, drug | DEAB | Sigma-Aldrich | D86256 | |
| Chemical compound, drug | All-trans retinoic acid (RA) | Sigma-Aldrich | R2625 | |
| Chemical compound, drug | Ketoconazole | Tocris | Tocris#1103 | |
| Chemical compound, drug | Cycloheximide | Sigma-Aldrich | C4859 | |
| Chemical compound, drug | Dexamethasone | Sigma-Aldrich | Sigma D4902 | |
| Peptide, recombinant protein | CAS9 | PNA Bio | CP01-20 | |
| Sequence-based reagent | *X. tropicalis tbx5* exon5 sgRNA | IDT DNA | GGGGTTCTGATATGAAGTGA | *Steimle et al., 2018* |
| Sequence-based reagent | *X. laevis* Tbx5 MO1 | GeneTools | 5'-TTA GGA AAG TGT CTC TGG TGT TGC C -3'; | *Brown et al., 2005* |

*Continued on next page*

*Continued*

| Reagent type (species) or resource | Designation | Source or reference | Identifiers | Additional information |
|---|---|---|---|---|
| Sequence-based reagent | *X. laevis* Tbx5 3 bp mismistach MO1 | GeneTools | 5'-TCA GTA AAG TAT CTC TGG TGT TGC C-3' | This paper |
| Sequence-based reagent | *X. laevis* Tbx5 MO2 | GeneTools | 5'-CAT AAG CCT CCT CTG TGT CCG CCA T-3 | ***Brown et al., 2005*** |
| Sequence-based reagent | *X. laevis* Tbx5 3 bp mismatch MO2 | GeneTools | 5'-TAT CAG ACT CCT CTG TGT CCG CCA T-3' | This paper |
| Sequence-based reagent | *X. laevis* Aldh1a2-MO | GeneTools | 5'-GCA TCT CTA TTT TAC TGG AAG TCAT-3' | ***Strate et al., 2009*** |
| Sequence-based reagent | *X. laevis* Cyp26a1-MO | GeneTools | 5'-TAG TGA GCA GAG TAT ACA GAT CCA T-3' | ***Janesick et al., 2013*** |
| Sequence-based reagent | *X. laevis* Cyp26c1-MO | GeneTools | 5'-TAC AAG ATG TTC CTC CTT GAG ATC A-3' | ***Yu et al., 2016*** |
| Commercial assay, kit | Protein G-conjugated magnetic beads | Life Technologies | 1,003D | |
| Commercial assay, kit | NEBNext Ultra DNA Library Prep Kit | New England Biolabs | E7370S | |
| Commercial assay, kit | Sera-Mag magnetic beads | GE | 6515-2105-050-250 | |
| Software, algorithm | Morpheus | Broad Institute | https://software.broadinstitute.org/morpheus RRID:SCR_017386 | |
| Commercial assay, kit | SceI mega-nuclease enzyme | New England Biolabs | R0694S | Use within 1 month of purchase, store at – 80°C |
| Commercial assay, kit | Dispase | Corning Life Sciences | 354235 | Use at 10 U/ml on *Xenopus* explants |

## *Xenopus* methods

### *Xenopus* embryo injections

WT adult *X. laevis* and *X. tropicalis* frogs were purchased from Nasco (Fort Atkinson, WI). Adult transgenic *X. laevis* and *X. tropicalis* Wnt/B-catenin reporter (*Xla.Tg(WntREs:dEGFP)*$^{Vlemx}$, NXR_0064; and *Xtr. Tg(WntREs:dEGFP)*$^{Vlemx}$, NXR_1094), and adult transgenic *X. laevis* nkx2-5:GFP (*Xla.Tg.(nkx2-5:GFP)*$^{Mohu}$, NXR_0030) frogs were purchased from the National *Xenopus* Resource (RRID:SCR_013713). Ovulation, in-vitro fertilization and natural mating, embryo de-jellying, and microinjection were performed as described (***Sive et al., 2000***). Plasmids for *Xenopus* GR-Tbx5 (Addgene 117248), *Xenopus* Tbx5 (Addgene 117247) (***Horb and Thomsen, 1999***), and *Xenopus* dominant-negative RARa (***Sharpe and Goldstone, 1997***) were previously described. Human *TBX5* (Horizon Discovery OHS5894-202500411) was gateway sub-cloned from its entry vector pENTR223 into the expression vector pCSf107mT-Gateway-3'myc (Addgene 67617) using clonase (ThermoFisher 11791020) according to manufacturer's instructions. Linearized plasmid templates were used to make mRNA for injection using the Ambion mMessage mMachine SP6 RNA Synthesis Kit (ThermoFisher AM1340). Total amounts of injected mRNA were as follows: GR-Tbx5 RNA, 125 pg; dN-RARa, 200 pg; and human TBX5-myc, 100 pg. Previously validated translation-blocking MOs against Tbx5 (***Brown et al., 2005***; ***Steimle et al., 2018***), Aldh1a2 (***Strate et al., 2009***), Cyp26a1 (***Janesick et al., 2013***), and Cyp26c1 (***Yu et al., 2016***) were injected at the 8-cell stage (for Tbx5-MO: a mixture of 2.5 ng each MO1 +2 per dorsal marginal zone (dmz) in *X. laevis*; mixture of 0.5 ng each MO1 +2 per dmz in *X. tropicalis*). MOs were purchased from GeneTools (Philmath, OR) and were as follows: Tbx5-MO1: 5'-TTA GGA AAG TGT CTC TGG TGT TGC C-3'; a negative control Tbx5 mismatch MO1 with three nucleotides mutated: 5'-T**C**A G**T**A AAG T**A**T CTC TGG TGT TGC C-3'; Tbx5-MO2: 5'-CAT AAG CCT CCT CTG TGT CCG CCA T-3'; Tbx5 3 bp mismatch MO2: 5'-**T**AT **C**AG **A**CT CCT CTG TGT CCG CCA T-3' (mismatch bases are indicated in bold, underlined); Aldh1a2-MO: 5'-GCA TCT CTA TTT TAC TGG AAG TCAT-3'; Cyp26a1 MO: 5'-TAG TGA GCA GAG TAT ACA GAT CCA T-3'; and Cyp26c1 MO: 5'-TAC AAG ATG TTC CTC CTT GAG ATC A-3'.

For F0 CISPR-mediated indel mutations, a sgRNA targeting *X. trop tbx5* exon 5 (DNA-binding domain) that causes frameshift mutations was synthesized in-vitro as previously described (*Steimle et al., 2018*). This exon5 sgRNA causes approximately 40% of injected embryos to have a phenotype (*Figure 2—source data 1*). Briefly, 2 nl of a mixture containing 50 pg/nl sgRNA with 0.5 ng/nl Cas9 protein (PNA Bio CP01-20) was injected on either side of the sperm entry point at the 1-cell stage (total of 200 pg sgRNA and 2 ng Cas9 protein per embryo).

For *Xenopus* whole embryo small-molecule treatments, embryos were cultured in 0.1× MBS +50 µg/ml gent with concentrations of 1 µM dexamethasone, 25 nM RA, or 10 µM DEAB (Sigma D86256). In all experiments, corresponding amounts of vehicle controls (DMSO or 0.2% fatty-acid free BSA) were used.

Gastrula or CP-foregut explants (containing both endoderm and lpm) were micro-dissected in 1× MBS +50 µg/ml gentamycin sulfate (gent; MP Biochemicals 1676045)±10 U/ml dispase (Corning Life Sciences 354235; to help remove the lpm) and were cultured in 0.5× MBS + 0.2% fatty acid free BSA (Fisher BP9704)+50 µg/ml gent with the following concentrations of factors: 1 µM dexamethasone (DEX; Sigma D4902); 1 µM cycloheximide (CHX; Sigma C4859); 25 nM all-trans RA (Sigma R2625); 100 ng/ml WNT2B (R&D Systems 3900-WN-025); 1 µM DEAB (Sigma D86256); 100 ng/ml FGF8b (R&D Systems 423-F8-025); and 0.5 µM ketoconazole (Tocris 1103). In CHX experiments, explants were treated for 2 hr in CHX prior to DEX+CHX treatment for 6 hr.

## *Xenopus* RT-qPCR

Xenopus explants were dissected from embryos of 2–3 separate fertilization/injection experiments, frozen on dry ice in 200 µl of TRIzol (ThermoFisher 15596018), and stored at − 80°C. RNA was extracted using TRIzol and purified using the Direct-zol RNA miniprep plus kit (ZymoResearch R2070); 500 ng RNA was used in cDNA synthesis reactions using Superscript Vilo Mastermix (ThermoFisher 11755050), all according to the manufacturer's instructions. qPCR reactions were carried out using PowerUp Mastermix (ThermoFisher A25742) on ABI StepOnePlus or QuantStudio3 machines. *Xenopus* RT-qPCR primer sequences are listed in *Supplementary file 1*. Relative expression, normalized to ubiquitously expressed *odc*, was determined using the $2^{-\Delta\Delta Ct}$ method. Graphs display the average $2^{-\Delta\Delta Ct}$ value ± standard deviation. Statistical significance (p<0.05) was determined using parametric two-tailed paired t-test, relative to uninjected, untreated explants. Each black dot in the RT-qPCR graphs represents an independent biological replicate containing four explants. Heat map of *Xenopus* RT-qPCR gene expression was generated using Morpheus software (https://software.broadinstitute.org/morpheus/) and shows the average $2^{-\Delta\Delta Ct}$ value from three biological replicates for each condition.

## *Xenopus* in-situ hybridization

In-situ hybridization of *Xenopus* embryos was performed as described (*Sive et al., 2000*) with minor modifications. Briefly, embryos were fixed overnight at 4°C in MEMFA (0.1 M MOPS, 2 mM EGTA, 1 mM MgSO4, and 3.7% formaldehyde), washed 3× 5 min in MEMFA without formaldehyde, dehydrated directly into 100% ethanol, washed 5–6 times in 100% ethanol, and stored at − 20°C for at least 24 hr. Proteinase K (ThermoFisher AM2548) on day 1 was used at 2 µg/ml for 10 min on stage NF15, NF25 embryos and 5 µg/ml on NF34 embryos; hybridization buffer included 0.1% SDS; RNAse A (ThermoFisher 12091021) used at 0.5 µg/ml; and anti-DIG-alkaline phosphatase antibody (Sigma 11093274910) used at 1:5000 in MAB buffer (100 mM Maleic acid, 150 mM NaCl, and pH 7.5) + 10% heat-inactivated lamb serum (Gibco 16070096) + 2% blocking reagent (Sigma 11096176001). Anti-sense DIG-labeled in-situ probes were generated using linearized plasmid cDNA templates with 10× DIG RNA labeling mix (Sigma 11277073910) according to the manufacturer's instructions.

## *Xenopus* immunofluorescence

Embryos were fixed in 100 mM HEPES (pH 7.5), 100 mM NaCl, 2.7% methanol-free formaldehyde for 2 hr at room temperature, dehydrated directly into Dent's post-fixative (80% Methanol/ 20% DMSO), washed five times in Dent's, and stored in Dent's at − 20°C for at least 48 hr. Embryos were serially rehydrated (75%, 40%, 25% methanol) into PBS + 0.1% TritonX-100 (PBSTr). Embryos were then cut in a transverse plane through the pharynx and posterior to the liver to create a foregut sample using a fine razor blade on a 2% agarose-coated dish in PBSTr. Foreguts were subjected to antigen retrieval in 1× R-universal epitope recovery buffer (Electron Microscopy Sciences 62719-10) for 1 hr

at 60–65°C, washed 2× 10 min in PBSTr, blocked for 1–2 hr in PBSTr + 10% normal donkey serum (Jackson ImmunoResearch 017-000-001) + 1 1% DMSO at room temperature, and incubated over-night at 4°C in this blocking solution+primary antibodies: chicken anti-GFP (Aves GPF-1020; diluted 1:1000), mouse anti-Sox2 (Abcam ab79351; 1:1000), rabbit anti-Aldh1a2 (Abcam ab96060; 1:500), and goat anti-Tbx5 (Santa Cruz Biotechnology sc-17866, 1:350). Secondary antibodies were donkey anti-chicken 488, donkey anti-rabbit Cy3, and donkey anti-mouse Cy5, donkey anti-goat 405 (Jackson ImmunoResearch 703-546-155, 711-166-152, 715-175-151, and 705-476-147, respectively; all used at 1:1000 dilution). After extensive washing in PBSTr, samples were incubated overnight at 4°C in PBSTr + 0.2% DMSO+secondary antibodies. Samples were again extensively washed in PBSTr, dehy-drated into 100% methanol, washed five times in 100% methanol, cleared, and imaged in Murray's Clear (two parts benzyl benzoate, one part benzyl alcohol) on a metal slide with glass coverslip bottom using a Nikon A1R confocal microscope to obtain optical sections.

## *Xenopus* luciferase assays

The *Xenopus tropicalis* v9.1 genome on Xenbase.org; RRID:SCR_003280 (*Karimi et al., 2018*) was used to define *Xenopus* enhancers. For *Xenopus* luciferase assays, the sequences of the mouse *Aldh1a2* Enh1 (chr9:71241739–71242765; mm10 genome), *X. trop aldh1a2* enh1 (chr3:89631924–89632943; v9.1/xenTro9 genome), *X.trop shh* MACS1(chr6:9535614–9536245; v9.1/xenTro9 genome), and human *SHH* MACS1 (chr7:156459384–156460049; hg19 genome) enhancers, as well as their respective mutant forms, were commercially synthesized (GenScript USA, Piscataway NJ; or IDT DNA, Coralville, IA) and cloned into the pGL4.23 firefly luc2/miniP vector (Promega E8411). For enh1 enhancer assays, embryos were co-injected with 5 pg of pRL-TK:renilla luciferase plasmid (Promega E2241) + 50 pg of the pGL4.23 luc2/miniP enhancer:luciferase plasmid and the following amounts of MOs or mRNAs into each dorsal marginal zone (dmz) region of 4–8 cell embryos: 3.5 ng of tbx5-MO or 3 bp mismatch-MO; 62.5 pg GR-Tbx5 RNA; 50 pg *Xenopus* Tbx5 RNA; 50 pg human TBX5-myc RNA. For hindgut mesendoderm injections, the luciferase reporters were injected into the ventral-posterior marginal zone at the 4–8 cell stage±Tbx5 RNA. For analysis of Shh MACS1 enhancer activity in endoderm, C1 (foregut) or C4 (hindgut) blastomeres were injected at the 16- or 32-cell stage with 5 pg pRL-TK +50 pg MACS1:luc±100 pg dnRARa RNA.

Each biological replicate contained a pool of five embryos, obtained from 2 to 3 separate fertil-ization/injection experiments which were frozen on dry ice in a minimal volume of 0.1× MBS and stored at – 80°C. To assay luciferase activity samples were lysed in 100 µl of 100 mM TRIS-Cl pH 7.5, centrifuged for 10 min at ~13,000×*g* and then 25 µl of the clear supernatant lysate was used separately in firefly (Biotium #30085-1) and renilla (Biotium 300821) luciferase assays according to the manufacturer's instructions. Relative luciferase activity was determined by normalizing firefly to renilla levels for each sample. Graph show the average relative luciferase activity ± standard deviation with dots showing values of biological replicates. Statistical significance was determined by parametric two-tailed paired t-test, *p<0.05.

## *Xenopus* transgenesis

Transgenesis was carried out using the I-SceI meganuclease procedure (*Ogino et al., 2006*; *Pan et al., 2006*; *Rankin et al., 2009* ). *Xenopus* transgenic plasmids were constructed using the pI-SceI-d2EGFP plasmid backbone (Addgene 32674). First, a fragment containing the mouse or *X. trop* enh1 enhancers upstream of a minimal TATA box promoter (*Tran et al., 2010*) flanked by duplicated copies of the 250 bp chick B-globin HS4 insulator (*Allen and Weeks, 2009*; *Rankin et al., 2011*) was commer-cially synthesized (GenScript USA) and cloned into the ApaI/XhoI sites of pBluescript II KS+ (Agilent 212207). ApaI/XhoI digestion released this fragment, and it was ligated into ApaI/XhoI digested pI-SceI-d2EGFP plasmid. The meganuclease reaction contained 200 ng DNA, 2.5 µl I-SceI enzyme (New England Biolabs R0694S; kept at – 80°C and used within 1 month of purchase) in 20 µl total volume and was incubated at 37°C for 30 min. 5 nl was then injected two times into 1 cell embryo on either side of the sperm entry point (10 nl total of meganuclease reaction injected per embryo). We observed 14/102 (~13%) and 21/183 (11%) GFP+full transgenic embryos using the mouse and *X. trop* enh1 constructs, respectively, from two independent injection experiments. As a negative control, 0/87 embryos were GFP positive when injected using reactions that omitted the I-SceI enzyme.

## Mouse methods

### RNA-seq

RNA-seq of the micro-dissected E9.5 WT and $Tbx5^{-/-}$ pSHF/CPP was previously published and is available on GEO (*Steimle et al., 2018*, GSE75077). Heat maps were generated using Morpheus software (https://software.broadinstitute.org/morpheus/). Columns in the *Figure 1* heat map represent biological replicates ($Tbx5^{+/+}$ WT N=5, $Tbx5^{-/-}$ N=2), and each column replicate contained n=4 pooled CP dissected regions.

### RT-qPCR, in-situ hybridization, and immunofluorescence

RT-qPCR of dissected, pooled (n=4) mouse E9.5 pSHF/CPP regions was performed as described (*Steimle et al., 2018*), cDNA generated using SuperScript III First-Strand Synthesis SuperMix (ThermoFisher 18080051), and qPCR was performed using PowerUp Mastermix (ThermoFisher A25742). Gene-expression levels were normalized by *Gapdh* and RT-qPCR primers are listed in *Supplementary file 1*. In-situ hybridization on mouse embryos was performed as described (*Hoffmann et al., 2009*). *Shh* probe was provided by Elizabeth Grove (University of Chicago). Immunofluorescence of WT CD-1 (Charles River) or *Shh:GFP* (B6.129 × 1(Cg)-*Shh*$^{tm6Amc}$/J; Jax Labs Stock Number #008466) mouse embryos was performed as described (*Rankin et al., 2016*; *Rankin et al., 2018*) using mouse anti-Aldh1a2 (Santa Cruz Biotechnology sc-166362; 1:1000), goat anti-Tbx5 (Santa Cruz Biotechnology sc-17866, 1:300), rabbit anti-Nkx2-1 (Santa Cruz Biotechnology sc-13040 H-190, 1:500), and chicken anti-GFP (Aves Labs GFP-1020, 1:1000).

Reconstructions of whole-mount in-situ hybridizations were generated using previously published methods (*Steimle et al., 2018*). In brief, images were obtained and pre-processed using Adobe Photoshop CS3 Extended (version 10.0.1, http://www.adobe.com) and reconstructed with AMIRA (version 5.3.2, http://www.amira.com). Manual review of each image in the stack was performed and corrections were made when necessary. LabelFields for gene expression and tissue were generated from the same series of sections using separate CastField and LabelVoxel modules. The SurfaceGen module was used to generate surfaces from these LabelFields. Gene expression models for two different genes were initially aligned using the Landmark (two sets) module, and a minimum of three landmarks were used to align the separate models. These landmarks were located using the pharyngeal endoderm and ventral edge of the SHF. Final alignments were fine-tuned manually using the Transform editor.

### Digital in-situ hybridization

Digital in-situ hybridization for *Aldh1a2*, *Tbx1*, *Tbx5*, and *Shh* was generated using the single cells Spatial Mouse Atlas (*Lohoff et al., 2021*; https://crukci.shinyapps.io/SpatialMouseAtlas/). Images are centered around the cardiothoracic region with cardiomyocytes highlighted.

### Mouse ESCs

The inducible *Tbx5*OE-mESC line was previously generated (*Steimle et al., 2018*) and differentiated to the cardiac lineage as described (*Kattman et al., 2011*). Doxycycline (Sigma D9891; concentrations of 0, 5, 10, 25, 50, 100, 250, and 500 ng/ml) was applied at the cardiac progenitor-like stage (day 6) and cells were harvested for RNA 24 hr later.

### ChIP-seq

ChIP-seq was performed using dissected whole lungs from E14.5 CD-1 mouse embryos obtained from Charles River. Chromatin was prepared as previously described (*Steimle et al., 2018*). For immunoprecipitation, the chromatin extract was incubated with 5 µg of the anti-TBX5 antibody (Santa Cruz Biotechnology sc-17866; Lot #G1516) at 4°C for >12 hr in a total volume of 200 µl. The immune complexes were captured by Protein G-conjugated magnetic beads (Life Technologies, 1003D) and washed as previously described (*Steimle et al., 2018*). The captured chromatin was eluted in ChIP Elution Buffer (10 mM Tris-HCl, pH 8.0, 1 mM EDTA, 1% SDS, and 250 mM NaCl) at 65°C. After RNase and proteinase K treatment and reverse cross-linking, DNA was purified. High-throughput sequencing libraries from ChIP and input DNA were prepared using NEBNext Ultra DNA Library Prep Kit (New England Biolabs, E7370S). During library preparation, adaptor-ligated DNA fragments

of 200–650 bp in size were selected before PCR amplification using Sera-Mag magnetic beads (GE, 6515-2105-050-250). DNA libraries were sequenced using Illumina Hi-seq instruments (single-end 50 base) by the Genomics Core Facility at the University of Chicago.

## Bioinformatics

### ChIP-seq analysis

Raw sequencing reads were aligned to the mm10 genome using Bowtie2 (*Langmead and Salzberg, 2012*) and SAMtools (*Li et al., 2009*) requiring a minimum mapping quality of 10 (−q 10). Pooled peak calling was performed using default settings of MACS2 callpeak (*Zhang et al., 2008*) with a q-value set to 0.05 and tag size set to 6 (−q 0.05 s 6). A fold-enrichment track was generated using MACS2 with the bdgcmp function (−m FE) for visualization on the IVG genome browser (*Thorvaldsdottir et al., 2012*). Public data reanalyzed in this study was downloaded from GEO either as Bigwig files or raw reads which were processed as described above.

### RNA-seq analysis

RNA-seq of the micro-dissected E9.5 WT and *Tbx5*$^{−/−}$ pSHF/CPP was previously published and is available on GEO (*Steimle et al., 2018*, GSE75077). This RNA-seq data was re-analyzed using Computational Suite for Bioinformaticians and Biologists (CSBB – v3.0.0) using *ProcessPublicData* module (https://github.com/praneet1988/Computational-Suite-For-Bioinformaticians-and-Biologists), (*Chaturvedi, 2019*). Differentially expressed genes (DEGs) between *Tbx5*$^{−/−}$ and WT were identified using RUVSeq, with a threshold of 1.5 fold change and 5% FDR. Expression heat maps were generated using Morpheus (https://software.broadinstitute.org/morpheus/).

DEGs were compared with gene sets from single-cell RNA-seq defining aSHF versus pSHF (*de Soysa et al., 2019*, GSE126128) and pharynx versus CPP+ lung progenitor cells (*Han et al., 2020*, GSE136689) from the early mouse embryo. We created a cardio-pharyngeal enriched gene set by combining marker genes of aSHF and pharynx mesendoderm, and a CP gene set by combining markers pSHF, pulmonary mesoderm, and lung endoderm. Overlaps in gene sets were visualized by Venn diagrams and significant overlaps were defined by HGTs. In addition, we assessed the enrichment of upregulated and downregulated DEGs from the *Tbx5*$^{−/−}$ embryos compared to the single-cell data sets by GSEA (*Subramanian et al., 2005*).

## Acknowledgements

The authors thank members of the Zorn and Wells labs for suggestions and Lisa Sandell (University of Louisville) for discussions. The authors thank Nicole Edwards and Leslie Brown for help generating and genotyping *Shh:GFP* mouse embryos.

## Additional information

### Funding

| Funder | Grant reference number | Author |
| --- | --- | --- |
| Eunice Kennedy Shriver National Institute of Child Health and Human Development | P01HD093363 | Aaron M Zorn |
| National Heart, Lung, and Blood Institute | R01HL092153 | Ivan P Moskowitz |
| National Heart, Lung, and Blood Institute | R01HL124836 | Ivan P Moskowitz |
| National Institute of General Medical Sciences | T32GM007183 | Jeffrey D Steimle |
| National Heart, Lung, and Blood Institute | T32HL007381 | Jeffrey D Steimle Ariel B Rydeen |

| Funder | Grant reference number | Author |
|--------|------------------------|--------|

The funders had no role in study design, data collection and interpretation, or the decision to submit the work for publication.

## Author contributions

Scott A Rankin, Conceptualization, Formal analysis, Investigation, Methodology, Validation, Visualization, Writing - original draft, Writing – review and editing; Jeffrey D Steimle, Conceptualization, Formal analysis, Investigation, Visualization, Writing – review and editing; Xinan H Yang, Ariel B Rydeen, Formal analysis, Investigation; Kunal Agarwal, Formal analysis, Software; Praneet Chaturvedi, Formal analysis, Software, Visualization, Writing – review and editing; Kohta Ikegami, Formal analysis, Investigation, Writing – review and editing; Michael J Herriges, Formal analysis, Investigation, Visualization; Ivan P Moskowitz, Funding acquisition, Project administration, Supervision, Writing – review and editing; Aaron M Zorn, Conceptualization, Funding acquisition, Project administration, Supervision, Writing – review and editing

## Author ORCIDs

Scott A Rankin ⓘ http://orcid.org/0000-0003-3953-6016
Michael J Herriges ⓘ http://orcid.org/0000-0002-7842-9515
Ivan P Moskowitz ⓘ http://orcid.org/0000-0003-0014-4963
Aaron M Zorn ⓘ http://orcid.org/0000-0003-3217-3590

## Ethics

Mouse and Xenopus experiments were performed according to Institutional Animal Care and Use Committee (IACUC) protocols (University of Chicago protocol 71737; Cincinnati Children's Hospital protocol 2019-0053).

## Decision letter and Author response

Decision letter https://doi.org/10.7554/eLife.69288.sa1
Author response https://doi.org/10.7554/eLife.69288.sa2

---

# Additional files

## Supplementary files

• Supplementary file 1. RT-qPCR primers used in this study for *Xenopus* and mouse.

• Supplementary file 2. Gene sets, utilized in *Figure 1*, from *de Soysa et al., 2019* (GSE126128) single-cell RNA-seq studies of developing E7.75-E9.5 mouse embryos that transcriptionally define the anterior and posterior second heart field territories.

• Supplementary file 3. Gene sets, utilized in *Figure 1*, from *Han et al., 2020* (GSE136689) single-cell RNA-seq studies of developing e8.75-e9.5 mouse embryos that transcriptionally define the cardiopulmonary progenitor (CPP)+ lung and ventral pharynx territories.

• Transparent reporting form

## Data availability

ChIP-seq data generated in this study is available from the Gene Expression Omnibus (GEO) accession number GSE167207.

The following dataset was generated:

| Author(s) | Year | Dataset title | Dataset URL | Database and Identifier |
|-----------|------|---------------|-------------|-------------------------|
| Steimle JD, Ikegami K, Burnicka-Turek O, Moskowitz IP | 2021 | TBX5 ChIP from the Fetal Lung | https://www.ncbi.nlm.nih.gov/geo/query/acc.cgi?acc=GSE167207 | NCBI Gene Expression Omnibus, GSE167207 |

The following previously published datasets were used:

| Author(s) | Year | Dataset title | Dataset URL | Database and Identifier |
|---|---|---|---|---|
| Yang XH, Steimle JD, Kweon J, Moskowitz IP | 2015 | Whole genome transcriptomic experiments in mice heart development system | https://www.ncbi.nlm.nih.gov/geo/query/acc.cgi?acc=GSE75077 | NCBI Gene Expression Omnibus, GSE75077 |
| de Soysa TY, Gifford CA, Srivastava D | 2019 | Single-cell analysis of cardiogenesis reveals basis for organ level developmental defects | https://www.ncbi.nlm.nih.gov/geo/query/acc.cgi?acc=GSE126128 | NCBI Gene Expression Omnibus, GSE126128 |
| Han L, Chaturvedi P, Zorn AM | 2020 | Single cell sequencing of dissected mouse foreguts at embryonic day 8.5 to 9.5 | https://www.ncbi.nlm.nih.gov/geo/query/acc.cgi?acc=GSE136689 | NCBI Gene Expression Omnibus, GSE136689 |
| Steimle JD, Ikegami K, Burnicka-Turek O, Moskowitz IP | 2020 | TBX5 ChIP from the Fetal Heart | https://www.ncbi.nlm.nih.gov/geo/query/acc.cgi?acc=GSE139803 | NCBI Gene Expression Omnibus, GSE139803 |
| Nicholas V, Sander M | 2020 | LSD1-mediated enhancer silencing enables endocrine cell development through attenuation of retinoic acid signaling | https://www.ncbi.nlm.nih.gov/geo/query/acc.cgi?acc=GSE104840 | NCBI Gene Expression Omnibus, GSE104840 |
| Wang A, Yue F Li Y | 2015 | Developmental Competence Encoded at the Level of Enhancers | https://www.ncbi.nlm.nih.gov/geo/query/acc.cgi?acc=GSE54471 | NCBI Gene Expression Omnibus, GSE54471 |

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
