## [Decision Letter]

**Acceptance summary:**

Your study provides critical data on the early events regulated by mesoderm-endoderm signaling required for lung development. These studies are important for assessment of early foregut development and help to define causal cell types in genetic diseases of lung formation.

**Decision letter after peer review:**

Thank you for submitting your article "TBX5 drives Aldh1a2 expression to regulate a RA-Hedgehog-Wnt gene regulatory network coordinating cardiopulmonary development" for consideration by *eLife*. Your article has been reviewed by 3 peer reviewers, and the evaluation has been overseen by a Reviewing Editor and Edward Morrisey as the Senior Editor. The following individual involved in review of your submission has agreed to reveal their identity: Tien Peng (Reviewer #1).

Essential revisions:

1) Please address the issues raised by the reviewers on the specificity of the pharmacological inhibitors through the use of more specific techniques such as morpholinos. Also clearly state the limitations of such pharmacological inhibitors.

2) Address the requests of additional histological and gene expression analysis with new data where noted by the reviewers.

3) The manuscript needs to be revised to better link the current studies with previous work and to highlight to the uninitiated the importance of your study.

4) The manuscript also needs additional revision to temper some of the claims as mentioned by the reviewers.

*Reviewer #1 (Recommendations for the authors):*

1. In Figure 1H,I, labeling of major anatomic landmarks would be helpful to delineate the domain of Aldh1a2 expression. Furthermore, it would be nice to highlight the relationship of the Shh expression domain alongside Aldh1a2 in Figure 1I as their complement anatomic relationship sets up the hypothesis studied in this manuscript.

2. In Figure 2a-c, it is clear that while there are some overlaps between Tbx5 and Aldh1a2, there are clearly domains of Aldh1a2 expression that are Tbx5-negative. It would be easier to visualize the overlap of the two domains in Figure 2b by coloring Tbx5 and Aldh1a2 green and red respectively with some quantification of the overlap. Furthermore, it is not clear how the reduction in Aldh1a2 is quantified. Finally, some discussion on what is driving Aldh1a2 expression in cells that are not Tbx5+.

3. In Figure 3A, a set of data shows that RA inhibition by DEAB can phenocopy RNA targeting of Tbx5, however, this is not really specific to the expression of Aldh1a2 per say. What is the phenotype of using MO to target Aldh1a2 in these set of experiments?

4. In Figure 4G, it would be nice to colocalize the expression of the GFP with in situ of Adh1a2 to see where the enhancer-driven GFP overlap with actual gene expression.

5. In Figure 5, I would raise a similar point as Comment 3, which is that the use of ketoconazole as a Cyp inhibitor is not as specific as MO targeting Cyp26a1. Would targeting of Cyp26a1 block the effect of FGF8 directly. This would provide stronger mechanistic proof for the model described.

6. In Figure 6E, can RA administration directly increase the reporter activity driven by the MACS1 enhancer? This would better link the Shh enhancer described to an RA response.

*Reviewer #2 (Recommendations for the authors):*

Prior to publication, a few points need to be considered.

1. Figure 1H. It will be nice to zoom in the foregut region in addition to showing the whole embryo. Although it is not necessary, it will be informative if the authors can show the in situ hybridization of Fgf10, Shh in TBX5 mutants using the similar projection method.

2. Fgf10 is known (also shown in Fig1H) to be localized in the future lung budding sites. In between the budding sites Fgf10 is not expressed. Is Tbx5 expressed in this region? It will be informative if authors can also show the projection image of Tbx5 as other genes in H.

3. Figure 1I, Can the authors label the projection images, e.g. heart field, lung budding site.

4. Does TBX5 deletion/kd induce stronger foregut phenotypes in frog than mouse? It appears that trachea-esophageal separation is abnormal in frog models (Figure 2B-C). Can the authors provide more details?

5. Does addition of exogenous Tbx5 RNA or RA in the rescue experiments affect the development of other organs? The presence of any phenotypes can further confirm that the extra Tbx5 or RA works in addition to the qPCR data.

*Reviewer #3 (Recommendations for the authors):*

Phrasing throughout the manuscript should be cautious. For example, differentially expressed genes are not necessarily Tbx5-dependent as worded on page 8 second paragraph.

Listing Tbx5 regulation of shh signaling as a key finding is an overstatement.

Can the authors demonstrate loss of Aldh1a2 expression in Tbx5null embryos?

The authors should mention the role of Aldh2a2 in retinoic acid processing so that the reader can understand how it functions to regulate RA signaling. The only description present is RA-producing enzyme.

The Wnt GFP reporter data in figure 2 B and C could be moved to supplemental data as it is more distracting than informative.

A co-stain for lateral plate mesoderm would be beneficial in figure 2 B and C.

For comparison purposes and because the readout is qualitative Figure 3 should be organized in a manner such that DEAB is next to tbx5 MO and 25 nM RA is next to mismatch MO. TBX5 RNA could go to the far right.

Limitation of drug inhibition should be stated. There are likely many more developmental processes disrupted by addition of RA inhibitor at the NF20 stage.

Authors might consider highlighting what new data they have added to the illustration in figure 7.

Figure legend 3 should be revised to reflect the panels in the figure.

Several of the effects observed in Figure 3 A-J appear to span beyond the cardiopulmonary region of the embryo. The author should recognized this in the interpretation of their data.

The diagram in 3L is based primarily on one time point examination. It may be prudent to remove this figure.

Wording in figure 4A legend has mixed verb tenses.

The statement that Tbx5 activates and represses certain genes goes beyond the data demonstrated in Figure 4A.

The authors should try to separate the description of the results from their discussion of the results. For example, p 12 the entire first full paragraph is more discussion than results.

---

## [Author Response]

Essential revisions:1) Please address the issues raised by the reviewers on the specificity of the pharmacological inhibitors through the use of more specific techniques such as morpholinos. Also clearly state the limitations of such pharmacological inhibitors.

We have repeated the Aldh1a2 and Cyp loss of function experiments with morpholinos that completely support the inhibitor studies. This data is presented in new Figure 3—figure supplement 1 and Figure 5—figure supplement 1. We have also clearly stated the limitations and advantages of pharmacological inhibitors.

2) Address the requests of additional histological and gene expression analysis with new data where noted by the reviewers.

We have added new analysis gene expression analysis in both sections and wholemount as requested and better annotated the anatomical structures in the images. These new data are presented in a revised Figure 1 D-F and in a revised Figure 1—figure supplement 1F-H.

2) The manuscript needs to be revised to better link the current studies with previous work and to highlight to the uninitiated the importance of your study.

We have revised the introduction and discussion to better link our results to previous work and to better highlight the importance of the work and the significance of the advance.

3) The manuscript also needs additional revision to temper some of the claims as mentioned by the reviewers.

We have revised the text to temper some of the claims as suggested by the reviewers.

Reviewer #1 (Recommendations for the authors):1. In Figure 1H,I, labeling of major anatomic landmarks would be helpful to delineate the domain of Aldh1a2 expression. Furthermore, it would be nice to highlight the relationship of the Shh expression domain alongside Aldh1a2 in Figure 1I as their complement anatomic relationship sets up the hypothesis studied in this manuscript.

In retrospect, we agree that the relative expression domains were not as clear as they should have been. While the each of these gene expression patterns has been documented before in previous publications, their direct comparison is key for our hypothesis. Therefore, we performed additional 3D whole mount confocal imaging of the E9.5 mouse embryo simultaneously staining Tbx5, Aldh1a2, Shh-GFP and Nkx2-1 (new panels in Figure 1D-F), which clearly shows the co-expression of Tbx5 and Aldh1a2 in the pSHF adjacent to the Shh/Nk2-1 expressing pulmonary endoderm. In addition, we used anew online single cells expression resource from the Marioni lab (Lohoff 2020, BioRxiv), to generate a “digital” in situs hybridization of sagittal section through the cardiothoracic region of a E9.5 mouse showing the overlapping expression domains of *Aldh1a2* and *Tbx5* in the pSHF, *Tbx1* in the aSHF and endodermal *Shh* (Figure 1—figure supplement 1F). We also added better labels of anatomical structures on the 3D in situ reconstructions as requested. Finally, we also present 3D maximum projections of confocal images showing the co-expression domain Tbx5-Ald1ha2 in the *Xenopus* pSHF/foregut lpm (new panel Figure 2B).

2. In Figure 2a-c, it is clear that while there are some overlaps between Tbx5 and Aldh1a2, there are clearly domains of Aldh1a2 expression that are Tbx5-negative. It would be easier to visualize the overlap of the two domains in Figure 2b by coloring Tbx5 and Aldh1a2 green and red respectively with some quantification of the overlap. Furthermore, it is not clear how the reduction in Aldh1a2 is quantified. Finally, some discussion on what is driving Aldh1a2 expression in cells that are not Tbx5+.

As requested, we revised the immunostaining in Figure 2B with Tbx5 in green and Aldh1a2 in red. This better shows the overlapping expression domains in yellow, which we quantified by generating a 3D volume with Imaris imaging software (new panel Figure 2B). We also quantified the reduction in Aldh1a2 protein levels based on the average immunostaining pixel intensity in the lpm in both the Tbx5-MO and tbx5-CRISPR mutants (New Figure 2—figure supplement 1B).

Yes there are some Aldh1a2 expression domains that do not overlap with Tbx5 including the somite and kidney mesenchyme (figures ??), which must be regulated by different enhancers independent of Tbx5. While studying these other enhancers and expression domains is interesting, this are not really relevant to the present and so we only briefly mention this in the revised discussion.

3. In Figure 3A, an set of data shows that RA inhibition by DEAB can phenocopy RNA targeting of Tbx5, however, this is not really specific to the expression of Aldh1a2 per say. What is the phenotype of using MO to target Aldh1a2 in these set of experiments?

We agree that this is an important point given the potential off target effects of inhibitors. As requested, we performed Aldh1a2-MO knockdowns with targeted microinjection to the foregut lpm. Immunostaining confirmed loss of Aldh1a2 protein, and resulted in gene expression changes largely identical to the DEAB treatment. Moreover, addition of exogenous RA rescued the Aldh1a2-MO phenotype. Since Aldh1a2 has earlier functions in the paraxial mesoderm, if the targeting was not efficient the embryos can have other defects, which is why we initially used the DEAB where we can control the timing of RA inhibition. We address the limitations and advantages of inhibitors in the revised manuscript. This new data presented in Figure 3—figure supplement 1A-E further strengthens our conclusions.

4. In Figure 4G, it would be nice to colocalize the expression of the GFP with in situ of Adh1a2 to see where the enhancer-driven GFP overlap with actual gene expression.

As requested, we performed co-staining of the enh1-GFP transgenes with endogenous Aldh1a2, which shows a specific overlap in the Tbx5+ foregut lpm, but not in the somites (new panels in Figure 4—figure supplement 1D-E). Again this is consistent with other Tbx5-independent enhancers regulating Aldh1a2 in other tissues.

5. In Figure 5, I would raise a similar point as Comment 3, which is that the use of ketoconazole as a Cyp inhibitor is not as specific as MO targeting Cyp26a1. Would targeting of Cyp26a1 block the effect of FGF8 directly. This would provide stronger mechanistic proof for the model described.

To address this, we performed MO knockdown of redundant Cyp26a1 and Cyp26a1, both of which are direct FGF targets. RT-PCR analysis of dissected foregut tissue shows that the cyp261a/c-MO phenocopies the Cyp-inhibitors ketoconazole, rescuing FGF8’s inhibition of *shh*, *tbx5*, *wnt2b* and *nkx2-1* expression and elevating their expression levels in untreated foregut explants (new Figure5—figure supplement 1). This new data provides a stronger mechanistic proof that FGF8 restricts RA-dependent cardiopulmonary development in the anterior foregut domain by regulating expression of Cyp26 RA-degrading enzymes. We thank the reviewer for the suggestion.

6. In Figure 6E, can RA administration directly increase the reporter activity driven by the MACS1 enhancer? This would better link the Shh enhancer described to an RA response.

We have added new luciferase assay data that demonstrate that both the human and *Xenopus* MACS1 enhancers can be activate by RA in isolated foregut endoderm and that this activation is dependent on RXR/RAR DNA-binding sites (new Figure 6—figure supplement 2).

Reviewer #2 (Recommendations for the authors):Prior to publication, a few points need to be considered.1. Figure 1H. It will be nice to zoom in the foregut region in addition to showing the whole embryo. Although it is not necessary, it will be informative if the authors can show the in situ hybridization of Fgf10, Shh in TBX5 mutants using the similar projection method.

As suggested, we have improved the imaging of the cardiopulmonary region with additional high magnification 3D whole mount confocal imaging of the E9.5 mouse embryo simultaneously staining Tbx5, Aldh1a2, Shh-GFP and Nkx2-1 (new panels in Figure 1D-F), which clearly shows the co-expression of Tbx5 and Aldh1a2 in the pSHF adjacent to the Shh/Nk2-1 expressing pulmonary endoderm. In addition, we used a new online single cells expression resource from the Marioni lab (Lohoff 2020, BioRxiv), to generate a “digital” in situs hybridization in sagittal section of the E9.5 mouse cardiothoracic region showing the overlapping expression domains of *Aldh1a2* and *Tbx5* in the pSHF, *Tbx1* in the aSHF and endodermal *Shh* (Figure 1—figure supplement 1F). We focused on Tbx5, Aldh1a2 and Shh rather than Fgf10 since these are the main focus of the work. As the mouse RNA-seq and RT-PCR as well as the *Xenopus* data show reduced Fgf10 and Shh expression of in Tbx5 mutants we did not repeat this by in situ in mice, given our time constraints.

2. Fgf10 is known (also shown in Fig1H) to be localized in the future lung budding sites. In between the budding sites Fgf10 is not expressed. Is Tbx5 expressed in this region? It will be informative if authors can also show the projection image of Tbx5 as other genes in H.

Arora et al., PLOS Genetics, 2012, have already published a description of Tbx5 and Tbx4 expression in the fetal mouse lung, including the distal mesenchyme where they regulate expression of *Fgf10*, *Wnt2*, as well as branching morphogenesis. It is unclear whether RA is involved in this role, but we have mentioned the similarity in the discussion.

3. Figure 1I, Can the authors label the projection images, e.g. heart field, lung budding site.

In addition to providing better analysis of the gene expression, we have labeled the anatomy on the 3D in situ reconstruction.

4. Does TBX5 deletion/kd induce stronger foregut phenotypes in frog than mouse? It appears that trachea-esophageal separation is abnormal in frog models (Figure 2B-C). Can the authors provide more details?

The mouse and *Xenopus* Tbx5 null mutant phenotypes are very similar. As previously described, both have major defects in cardiac morphogenesis and fail to specify respiratory fate in the ventral foregut (Bruneau et al., 2001 Cell, Brown et al., 2005 Development and Steimle et al., 2018 PNAS). Tbx5-/- null mice die between E9.5 and E10.5 due to cardiac defects, before trachea-esophageal separation at E11.5. *Xenopus* tadpoles on the other hand are well known to survive for many days with severe heart defects as they can absorb oxygen from the water, which is one reason that we used *Xenopus* for this study. We now make this clearer in the manuscript and show data on the late tbx5-MO phenotype confirming the previously published hypoplastic atria and vertical lacking trabeculae as well as the failed trachea-esophageal separation (new Figure 2—figure supplement 1).

5. Does addition of exogenous Tbx5 RNA or RA in the rescue experiments affect the development of other organs? The presence of any phenotypes can further confirm that the extra Tbx5 or RA works in addition to the qPCR data.

We did not look at other organs, but focused our analysis on early heart and lung specification, which is the theme of the paper. We show that addition of Tbx5 or RA can rescue cardiopulmonary development as assessed by immunostaining, in situ hybridization and RT-qPCR. There are publications suggesting that Tbx5 and RA may also interact in other tissue such as the limb, but the logic of the regulatory networks appears to be somewhat different. We address this in the discussion. While beyond the scope of this work, in the future it would be interesting to compare the different developmental contexts.

Reviewer #3 (Recommendations for the authors):Phrasing throughout the manuscript should be cautious. For example, differentially expressed genes are not necessarily Tbx5-dependent as worded on page 8 second paragraph.

We have revised the phasing to be more accurate and cautious as requested.

Listing Tbx5 regulation of shh signaling as a key finding is an overstatement.

We are not sure what the reviewer means here. In our view the finding that Tbx5 indirectly regulates *shh* expression via RA is a major finding and one of the key mechanisms linking mesoderm and endoderm development. We have revised how we describe this to be clear that Tbx5 regulates *shh* expression.

Can the authors demonstrate loss of Aldh1a2 expression in Tbx5null embryos?

The RNA-seq and RT-PCR of the Tbx5-null mouse foregut do clearly demonstrate the loss of Aldh1a2, as does the many experiments in *Xenopus*. Give the time constraints, and the other revisions requested did not repeat this experiment with immunostaining or in situ.

The authors should mention the role of Aldh2a2 in retinoic acid processing so that the reader can understand how it functions to regulate RA signaling. The only description present is RA-producing enzyme.

Yes, we agree that this would be helpful, and we have added this. Thank you for the suggestion.

The Wnt GFP reporter data in figure 2 B and C could be moved to supplemental data as it is more distracting than informative.A co-stain for lateral plate mesoderm would be beneficial in figure 2 B and C.

We understand the reviewer perspective, but we have shown the figure to a number of colleagues take a different view and find the Wnt-GFP reporter helpful as this shows the respiratory progenitors. Having said that we take the reviewers point and simplified the description of these data. We also show the panels without the GFP for those readers who find it distracting. To better show the Tbx5-Aldh1a2 co-staining in lateral plate mesoderm we have new immunostaining (Figure 2B) showing the overlap in the ventral-lateral foregut lateral plate mesoderm.

For comparison purposes and because the readout is qualitative Figure 3 should be organized in a manner such that DEAB is next to tbx5 MO and 25 nM RA is next to mismatch MO. TBX5 RNA could go to the far right.

We respectfully prefer to keep the original organization of the columns.

Limitation of drug inhibition should be stated. There are likely many more developmental processes disrupted by addition of RA inhibitor at the NF20 stage.

We have now clearly stated the limitation of the inhibitor experiments. In addition, we have added new Aldh1a2-MO and Cyp261a/c-MO experiments to complement the DEAB and ketoconazole experiments, which support our model. We also acknowledge that these inhibitors can affect other organ system notably the kidney where RA is known to play a role as published by others. While these to not change the interpretation in the cardiopulmonary tissue we agree that it is important to state this.

Authors might consider highlighting what new data they have added to the illustration in figure 7.

Thank you for this suggestion we agree that is helpful.

Figure legend 3 should be revised to reflect the panels in the figure.

Done. Thank you for pointing out this error.

Several of the effects observed in Figure 3 A-J appear to span beyond the cardiopulmonary region of the embryo. The author should recognized this in the interpretation of their data.

We agree that the Tbx5 depleted embryos have non-cell autonomous defects in other tissues including the kidney primordia, pharynx, head and ectoderm. We interpret this to be due to the disrupted RA and FGF signaling that we have documented. We have modified the text to explain this.

The diagram in 3L is based primarily on one time point examination. It may be prudent to remove this figure.

We have revised the figure legend of the model to make it clear that we are refereeing to a specific time point. We agree that this is important since Tbx5-RA interactions may be different at different times.

Wording in figure 4A legend has mixed verb tenses.

Thank you we have corrected this.

The statement that Tbx5 activates and represses certain genes goes beyond the data demonstrated in Figure 4A.

We have revise language to state that expression levels increase and decreases in Tbx5 GOF.

The authors should try to separate the description of the results from their discussion of the results. For example, p 12 the entire first full paragraph is more discussion than results.

We have done this, although we kept some description of previous work where is provides a rational for the experiments.